# On Scaled Normal Stresses in Multiaxial Fatigue and Their Exemplary Application to Ductile Cast Iron

**Michael Wächter** [1],*, **Alexander Linn** [1], **Ralf Wuthenow** [1], **Alfons Esderts** [1], **Christian Gaier** [2], **Jan Kraft** [3], **Carl Fällgren** [3] and **Michael Vormwald** [3]

1 Institute for Plant Engineering and Fatigue Analysis, Clausthal University of Technology, 38678 Clausthal-Zellerfeld, Germany; alexander.linn@tu-clausthal.de (A.L.); ralf.wuthenow@tu-clausthal.de (R.W.); alfons.esderts@tu-clausthal.de (A.E.)
2 Engineering Center Steyr Gmbh & Co. KG, 4300 St. Valentin, Austria; christian.gaier@magna.com
3 Materials Mechanics Group, Technical University of Darmstadt, 64287 Darmstadt, Germany; kraft@wm.tu-darmstadt.de (J.K.); faellgren@wm.tu-darmstadt.de (C.F.); vormwald@wm.tu-darmstadt.de (M.V.)
* Correspondence: michael.waechter@tu-clausthal.de

**Abstract:** The approaches used to calculate the fatigue life of components must inevitably consider multiaxial stresses. Compared to proportional loading, the calculation of nonproportional loading is particularly challenging, especially since different materials exhibit the effects of nonproportional hardening and shifts in fatigue life. In this paper, the critical plane approach of scaled normal stresses, first proposed by Gaier and Dannbauer and later published in a modified version by Riess et al., is investigated in detail. It is shown that, on the one hand, compatibilities exist or can be established with known proportional strength criteria that can account for the varying ductility of different materials. Furthermore, it is demonstrated that the scaled normal stress approach can be formulated in such a way that different strength criteria can be used therein. As an example, the generally formulated approach for scaled normal stresses is applied to test results from ductile cast iron material EN-GJS-500-14. Different correction factors accounting for nonproportional loading are investigated. Through appropriate parameterization of one of the studied corrections, proportional and nonproportional test results were observed to fall within one common scatter band.

**Keywords:** multiaxial fatigue; critical plane; scaled normal stresses; proportional loading; nonproportional loading; ductile cast iron

## 1. Introduction

Fatigue assessments for components relevant to safety are conducted either on an experimental basis or by using calculation approaches. From an economic point of view, calculation appears preferable at first glance. However, with regard to accuracy, the experimental approach is superior. Therefore, a combination of calculation and experiment is often used.

Service loads that are the reason for fatigue assessments usually consist of several components (e.g., different force and moment channels), which only behave proportionally to each other in special cases. Therefore, assessments of analytical fatigue strength need to be able to handle multiaxial nonproportional loads while producing reliable results.

This objective contrasts with the fatigue strength data commonly available in practice:

1. Strain–life curve: These Wöhler curves are determined in axial strain-controlled tests on unnotched material specimens [1,2], allowing the uniaxial stress state to be tested. As an alternative to experimental determination, these properties can also be estimated using statistical dependencies (see, for example, [3,4]). However, this does not change the range of validity of the properties.

2.   S–N curves of components or specimens: These Wöhler curves are determined in load-controlled fatigue tests on components or notched material specimens. Frequently, only one load component is applied, which leads to an approximately uniaxial or multiaxial proportional stress state within the component. As for strain–life curves, methods for the estimation of S–N curves exist, which lead to so-called synthetic S–N curves (e.g., [5]).

Apart from the properties described above, more complex fatigue tests can of course also be used to determine fatigue properties for shear loading or multiaxial nonproportional loading. Nevertheless, these tests cannot be taken for granted, since they are associated with high or at least higher costs and may make the analytical assessment economically unviable. It thus seems to be straightforward to also retain the properties of uniaxial or multiaxial proportional stresses as a basis for analytical fatigue assessments in cases of nonproportional loading.

It should be noted, however, that more complex and physically well-founded calculation concepts are available that achieve good accuracies [6,7]. The expensive acquisition of the required input data, however, often prevents the application of property-intensive concepts [8]. Another reason is the trend to expect fatigue life calculations to be a largely automated postprocessing of finite element analyses, in which a local fatigue life is calculated for each node of the finite element mesh. Despite the increase in computational resources, the algorithms for nonproportional problems often still exhibit excessively long computation times.

The algorithms for proportional loads are common and well accepted. Their simplicity makes them accessible for newcomers to the field of fatigue analysis. A compatibility between established approaches for proportional loads and those for nonproportional loads is therefore desirable. This gives the advantages that, on the one hand, the existing material properties for uniaxial loading described above can continue to be used and, on the other, no discontinuities arise in results when components with loads of different degrees of nonproportionality are calculated.

One approach that can meet the described requirements for the calculation of fatigue strength under nonproportional loading is the critical plane approach with scaled normal stresses. It was originally proposed by Gaier and Dannbauer [9,10] and was later adopted in modified form by Riess et al. [11,12]. One of the major issues in dealing with nonproportional loading is the occurrence of two effects:

1.   Nonproportional hardening occurs in certain materials and leads to higher stress amplitude for the same strain level (e.g., [13]).
2.   Additionally, fatigue tests on components or notched specimens show shifts in fatigue life compared to proportional loading [14].

Both effects seem to be connected to the material ductility and will be discussed in detail in Section 2.4.

While many studies of the effects of nonproportional loading can be found for a wide variety of materials, consideration of the effects in component and variable amplitude design has not yet resulted in conclusive resolution [14].

The purpose of this paper is to show that compatibility can be established between the scaled normal stress approach and different multiaxial fatigue criteria, which are well accepted but applicable only for proportional loading. The issue of the abovementioned effects under nonproportional loading will also be addressed. This will be achieved using the example of a ductile cast iron material.

In order to achieve these goals, multiaxiality criteria that are known from literature and engineering practice are presented in Section 2. For the approaches that are suitable for application for nonproportional loading, the focus is on the idea of scaled normal stresses. In Section 3 that follows, the two previously presented scaled normal stress approaches are compared, and references are made to the criteria for proportional loading. This leads to a generalized formulation of the scaled normal stress approach in Section 4. In order to be able to demonstrate the application of the approach on a specific example, Section 5

presents new test results on the ductile cast iron material EN-GJS-500-14 followed by the application of the scaled normal stress method to this material. Finally, the main findings are summarized, and open questions concerning the scaled normal stress method are addressed in Section 6.

The evaluation of accuracy, i.e., through comparisons of a sufficient amount of test results for proportional and nonproportional loading, is not part of this work.

## 2. Calculation Approaches for Uniaxial, Proportional, and Nonproportional Loading

Two calculation approaches have been established for performing analytical fatigue assessments. These are, on the one hand, stress-based concepts in which linear elastic material behavior is assumed. They are therefore only suitable for the finite-fatigue life regime (number of cycles to failure $N > 10^4$) and the region of the fatigue limit, in which plastic deformations are negligibly small. On the other hand, strain-based concepts are used, in which the material behavior is represented more realistically because plastic deformations are taken into account. Whereas stress-based concepts are based on Wöhler curves of components or notched material specimens, strain-based concepts are based on results from strain-controlled tests of unnotched material specimens. In contrast to the former, the latter do not include influences specific to the component design (notch influences, surface treatment, etc.).

The two concept variants differ in the calculation effort due to the differences in assumed material behavior. Since plastic deformations are taken into account in strain-based concepts, either complex finite element analyses with elastic–plastic material behavior or notch root approximation methods [15–17] must be used. The application of strain-based concepts is therefore more computationally intensive but, at the same time, the only well-accepted way to include the low cycle fatigue regime in component designs.

Both concepts usually share the need for defining damaging events, usually so called load cycles. These events are required to perform some sort of damage accumulation calculation for determining the fatigue life for variable amplitude loading from a Wöhler curve that itself is determined for constant amplitude loading. To identify the aforementioned cycles within the load series, some sort of cycle-counting method (e.g., rainflow counting) needs to be applied. Especially for nonproportional stresses, the application of cycle counting methods poses a problem insofar as the ratios in the stress tensor are constantly changing. Therefore, the application of cycle counting methods to different stress components leads to differing results. Proposals for multichannel implementations of rainflow counting have been suggested (e.g., [18–20]). Apart from that, it shall be mentioned that calculation approaches can also be found in the literature that are not based on load cycles as damage-relevant quantity, but instead use the incrementally calculated increase in plastic energy, e.g., according to Jiang [21] or Volkov et al. [22]. The advantage of such an approach is that the problems associated with the application of rainflow counting in the case of nonproportional loading can be circumvented.

In engineering practice, both stress- and strain-based concepts face the problem of having to verify complex stress states and notch situations using properties that are as easy to determine as possible. Therefore, strength hypotheses are required that allow the comparison of different multiaxial stresses with uniaxial or multiaxial proportionally determined fatigue strength values.

Below, well-established fatigue strength hypotheses for proportional stress states are presented. Due to the large number of available hypotheses, however, only a brief insight can be given for the multiaxiality of those that are nonproportional. The focus is on the scaled normal stress approach. In addition, approaches to account for effects under nonproportional loadings are described briefly.

### 2.1. Strength Hypotheses for Proportional Loading

For proportional loading, criteria are often used that were originally developed for evaluating static yielding but have been transferred to fatigue applications. Socie and

Marquis point out that this approach benefits from the fact that only uniaxial properties are sufficient for describing the fatigue behavior [13].

The three most commonly used criteria to calculate uniaxial equivalent stress amplitudes $\sigma_{eq,a}$ are as follows:

- The maximum principal stress criterion according to Rankine [23]:

$$\sigma_{eq,a,R} = \max(|\sigma_{a,1}|, |\sigma_{a,2}|, |\sigma_{a,3}|) \xrightarrow{\text{if } \sigma_1 \geq \sigma_2 \geq \sigma_3} \begin{cases} |\sigma_{a,1}| \text{ for } |\sigma_{a,1}| \geq |\sigma_{a,3}| \\[2mm] |\sigma_{a,3}| \text{ for } |\sigma_{a,3}| \geq |\sigma_{a,1}| \end{cases} \tag{1}$$

- The maximum shear stress criterion according to Tresca [24]:

$$\sigma_{eq,a,T} = \max(|\sigma_{a,1} - \sigma_{a,2}|, |\sigma_{a,2} - \sigma_{a,3}|, |\sigma_{a,3} - \sigma_{a,1}|) \xrightarrow{\text{if } \sigma_1 \geq \sigma_2 \geq \sigma_3} |\sigma_{a,3} - \sigma_{a,1}| \tag{2}$$

- The distortion strain energy or octahedral shear stress criterion according to von Mises [25]:

$$\sigma_{eq,a,M} = \sqrt{\frac{1}{2}\left[(\sigma_{a,1} - \sigma_{a,2})^2 + (\sigma_{a,2} - \sigma_{a,3})^2 + (\sigma_{a,3} - \sigma_{a,1})^2\right]} \tag{3}$$

Here, $\sigma_{a,1}$, $\sigma_{a,2}$, and $\sigma_{a,3}$ are the amplitudes of the principal stresses. Dowling points out that the amplitudes, which are always positive by nature, must be given a sign in this case [26]. For this purpose, one of the stress components to be used must be specified as a reference. The selection can be performed arbitrarily. If the extrema of the other stress components occur in phase with the reference component, all amplitudes must be assigned the same (e.g., positive) sign. If individual stress components occur 180° out of phase with the reference component, their amplitudes are assigned a negative sign. At first glance, such a case, in which individual stress components occur 180° out of phase to each other, seems to be caused by nonproportional loading. In fact, such cases occur, for example, in round bars under pure torsional loading. Here, two principal stresses with different signs occur. Therefore, the oscillating stress signals of the two components are 180° out of phase and, yet, both signals are proportional to each other.

For plane stress states ($\sigma_3 = 0$) the failure criteria of the three hypotheses are represented by the plots in Figure 1a. For triaxial stress states, the failure criteria may also be plotted (see Figure 2). The hypotheses in Equations (1)–(3) reproduce various ratios of fatigue strengths $f_{\tau/\sigma}$ for shear stresses $\tau_e$ and normal stresses $\sigma_e$.

$$f_{\tau/\sigma} = \frac{\tau_e}{\sigma_e} \tag{4}$$

Both strengths are either defined for the same number of load cycles or represent the endurance limits for the stress types at reversed loading. The corresponding hypotheses being as follows (Figure 1b):

- Rankine: $f_{\tau/\sigma} = 1$;
- von Mises: $f_{\tau/\sigma} = \frac{1}{\sqrt{3}} = 0.58$;
- Tresca: $f_{\tau/\sigma} = 0.5$.

$f_{\tau/\sigma}$ can be understood as a measure of the ductility of a material with $f_{\tau/\sigma} = 1$ meaning brittle behavior, while $f_{\tau/\sigma} = 0.5$ or $\frac{1}{\sqrt{3}}$ is found with ductile materials. This also explains that the three hypotheses are used for the corresponding material behavior: while Rankine is used for brittle materials, both von Mises and Tresca are used to describe ductile materials. Sines points out that it is difficult to select between Tresca's and von Mises' hypotheses on the basis of experimental results because of the small differences between

the two hypotheses. Therefore, the selection should rather be made according to more pragmatic aspects (e.g., the "convenience of the mathematical expression") [27].

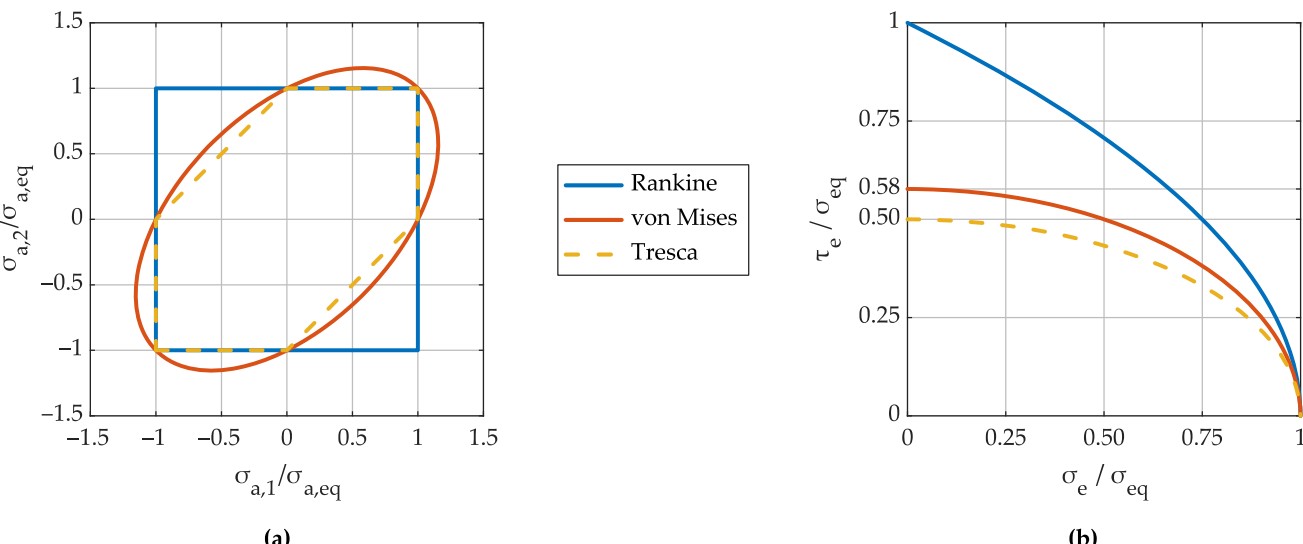

(a)

(b)

**Figure 1.** Visualization of the established strength hypotheses according to Rankine, von Mises, and Tresca for proportional loading as (**a**) principal stress diagram and (**b**) for different combinations of permutable normal and shear stress (both for plane stress states, $\sigma_3 = 0$.

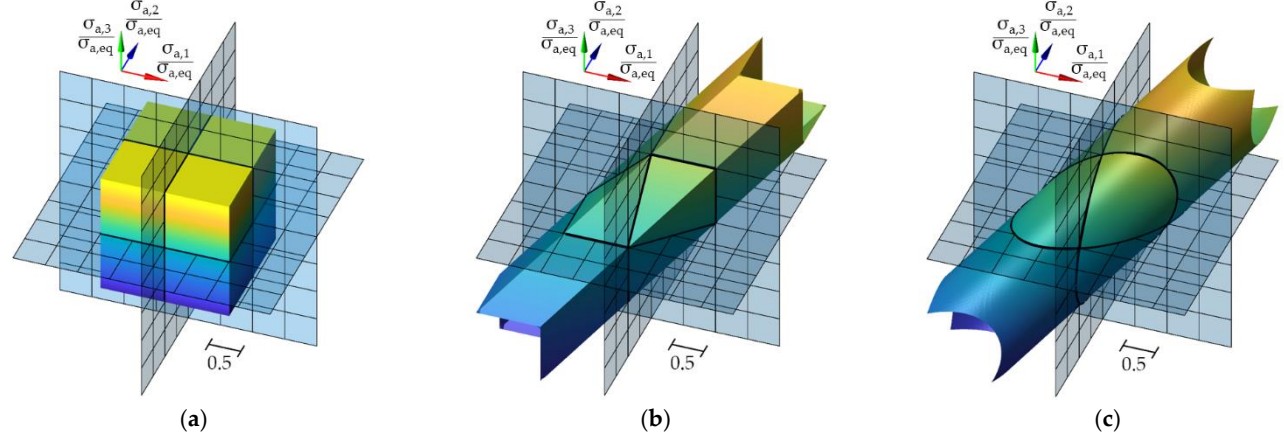

(a) (b) (c)

**Figure 2.** Visualization of the established strength hypotheses according to (**a**) Rankine, (**b**) Tresca, and (**c**) von Mises for proportional loading as principal stress diagrams for triaxial stress states.

While the behavior of either mainly brittle or mainly ductile materials may be described by the pairing Rankine and von Mises or Rankine and Tresca, different materials show values of $f_{\tau/\sigma}$ also between 0.5 and 1 [27]. The German FKM guideline [5] (Forschungskuratorium Maschinenbau), e.g., assigns the values given in Table 1 to certain metallic material groups.

While both steel and wrought aluminum may be accordingly described by the von Mises hypothesis and gray cast iron follows Rankine, the groups of other materials lie between the two hypotheses.

**Table 1.** General values for the ratio of endurance limits for shear and normal stresses according to FKM guideline [5].

| Material Group | $f_{\tau/\sigma}$ |
|---|---|
| wrought, forged and cast steel | $\frac{1}{\sqrt{3}} \approx 0.58$ |
| nodular graphite cast iron | 0.65 |
| malleable cast iron | 0.75 |
| gray cast iron | 1.00 |
| wrought aluminum | $\frac{1}{\sqrt{3}} \approx 0.58$ |
| cast aluminum | 0.75 |

El-Magd also points out that most materials that are used in engineering practice do not show the exact values of $f_{\tau/\sigma} = 0.5$, $\frac{1}{\sqrt{3}}$, or 1. He develops a strength hypothesis that is a linear combination of Equations (1) and (2) and, therefore, is able to describe materials that show values between $f_{\tau/\sigma} = 0.5$ and 1 [28]:

$$\sigma_{eq,a} = (1 - q) \cdot \sigma_{eq,a,T} + q \cdot \sigma_{eq,a,R} \tag{5}$$

Here, the quantity q controls the mixture between the two hypotheses:

$$q = 2 - \frac{1}{f_{\tau/\sigma}} = \frac{2 - \frac{1}{f_{\tau/\sigma}}}{2 - 1} \text{ with } (0.5 \leq f_{\tau/\sigma} \leq 1) \tag{6}$$

which results in the Tresca criterion for $f_{\tau/\sigma} = 0.5$ and in Rankine for $f_{\tau/\sigma} = 1$. It should be noted that El-Magd derives Equation (5) using test results in the range of the endurance limit and at plane stress states. However, there is no formal requirement that would prevent its use for triaxial stress states. For exemplary values of $f_{\tau/\sigma}$, the failure criteria are plotted in Figure 3. For triaxial stress states and $f_{\tau/\sigma} = 0.5$, the results are shown in Figure 2b, and the results for $f_{\tau/\sigma} = 1$ are shown in Figure 2a.

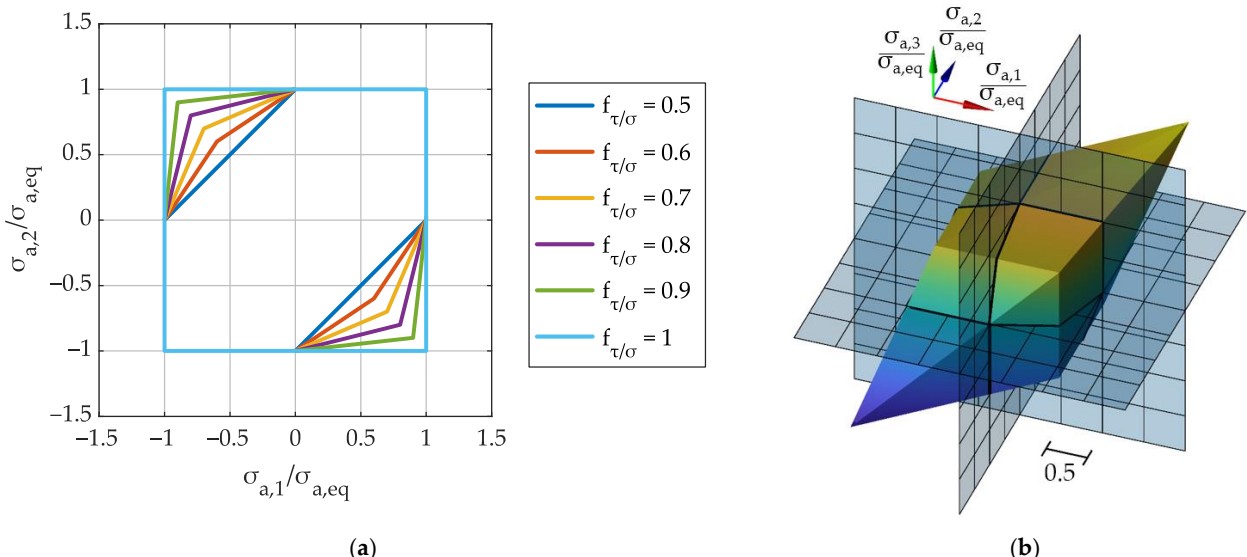

(**a**)  (**b**)

**Figure 3.** Strength hypothesis according to El-Magd for (**a**) different values for $f_{\tau/\sigma}$ and plane stress states, $\sigma_3 = 0$, and (**b**) $f_{\tau/\sigma} = 0.75$ and triaxial stress states.

Lüpfert and Spies transfer the idea of a linear combination of two strength hypotheses to the pair of hypotheses of Rankine and von Mises [29,30]:

$$\sigma_{eq,a} = (1 - q) \cdot \sigma_{eq,a,M} + q \cdot \sigma_{eq,a,R} \tag{7}$$

$$q = \frac{\sqrt{3} - \frac{1}{f_{\tau/\sigma}}}{\sqrt{3} - 1} \text{ with } \left( \frac{1}{\sqrt{3}} \leq f_{\tau/\sigma} \leq 1 \right) \tag{8}$$

For exemplary values of $f_{\tau/\sigma}$, the failure criteria are plotted in Figure 4. For triaxial stress states and $f_{\tau/\sigma} = 0.58$, the results are shown in Figure 2c, and the results for $f_{\tau/\sigma} = 1$ are shown in Figure 2a. The mixed strength hypotheses of Equations (7) and (8) are used and widely accepted in the German standard of the FKM guideline [5].

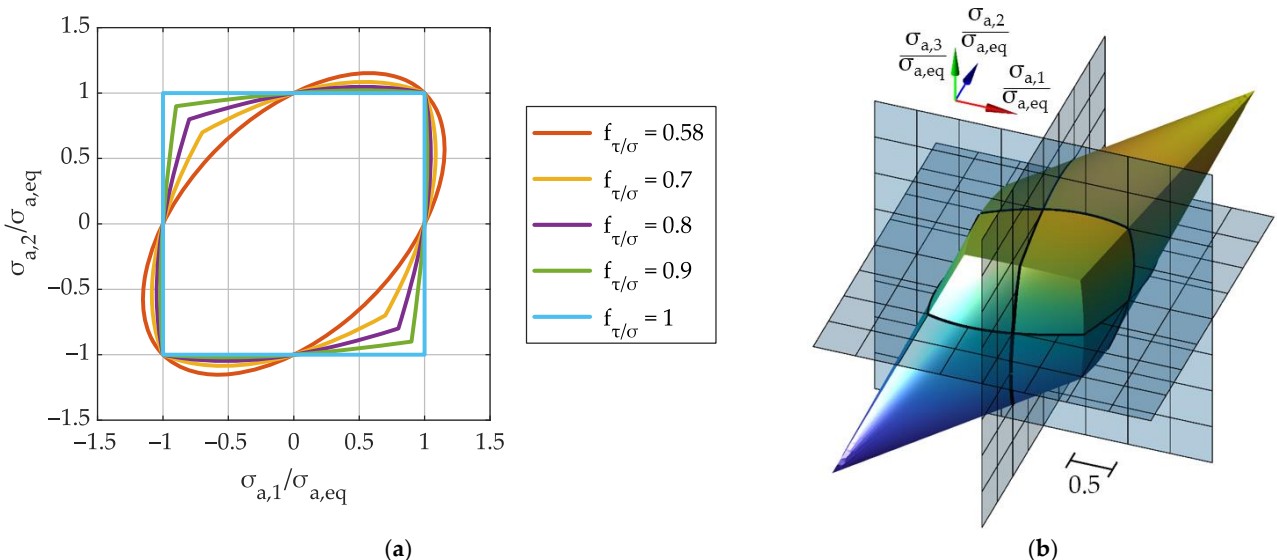

(a)                                                                                                            (b)

**Figure 4.** Strength hypothesis according to Lüpfert and Spies for (**a**) different values for $f_{\tau/\sigma}$ and plane stress states, $\sigma_3 = 0$, and (**b**) $f_{\tau/\sigma} = 0.75$ and triaxial stress states.

### 2.2. Time Sequences of Equivalent Stresses

The strength hypotheses presented in the previous section are formulated using stress amplitudes. This requires the identification of the individual load cycles before the strength hypotheses are applied. The reason for this can be found in the historical development of the hypotheses, which were initially primarily applied in stress-based concepts for assessments of the endurance limit. In this case, only knowledge of the largest occurring stress amplitude is necessary. Later on, these hypotheses were adapted for fatigue assessments in the finite fatigue life regime in conjunction with Miner's rule. In these cases, too, the stress amplitudes may be used as an input parameter for the strength hypotheses (e.g., [5]). In the case of proportional loading, cycle counting for each of the components of a given stress tensor would result in identical load cycles.

An alternative procedure, however, can be found, for example in [31–35]. Instead of applying the strength hypotheses to the stress amplitudes, the time sequences of the stress tensor components are used to generate a one-component equivalent stress–time series; i.e., a single value of the equivalent stress is calculated for each entry of the time series. The equivalent stress–time series can be fed into a cycle counting algorithm, e.g., rainflow counting. This results in amplitudes and mean values of equivalent stresses, which can be introduced into mean stress correction and, after that, into damage calculation. If one of the Equations (1)–(7) is used to calculate the equivalent stress–time series, the obvious problem would arise that regardless of the stress state present, positive values would always result for the equivalent stresses. This cannot be appropriate from a physical point of view and will lead to small stress amplitudes and only tensile mean stresses. Therefore, in addition to the equivalent stress value, a sign must be defined to indicate whether the stress state is tensile or compressive [31].

To define a sign for the equivalent stress, Bishop and Sherratt [31] suggested using the sign of the principal stress with the largest absolute value, Equation (9).

$$\sigma_{eq,signed} = \begin{cases} \sigma_{eq} & \text{for } |\sigma_1| \geq |\sigma_3| \\ -\sigma_{eq} & \text{for } |\sigma_1| < |\sigma_3| \end{cases} \tag{9}$$

It should be noted that Equation (9) is only valid if applying the often-used convention of sorting the principal stresses according to their magnitude so that $\sigma_3 \leq \sigma_2 \leq \sigma_1$.

Others propose using the sign of the first invariant $I_1$ of the stress tensor, Equation (10) [34]. This is equivalent to the sign of the hydrostatic stress, which is used, e.g., by Zhang et al. [35].

$$\sigma_{eq,signed} = \text{sign}(I_1) \cdot \sigma_{eq} = \text{sign}(\sigma_1 + \sigma_2 + \sigma_3) \cdot \sigma_{eq} \tag{10}$$

Contrary to the frequently encountered definition of the sign function, where

$$\text{sign}(0) = 0 \tag{11}$$

here

$$\text{sign}(0) = 1 \tag{12}$$

is assigned since, otherwise, due to the sign function, an equivalent stress of 0 would result for stress states that clearly represent a loaded component, e.g., pure torsion with $\sigma_2 = -\sigma_1$ and $\sigma_3 = 0$.

Many applications show a plane stress state in the critical area if the latter is located at the component surface and if the surface is not loaded by pressure or friction forces. In this case of plane stress state (e.g., $\sigma_3 = 0$), the signs given by Equations (9) and (10) are identical. Only in the case of triaxial stress state may different signs occur.

As described, Equations (9) and (10) can be used to add a sign to each of the hypotheses in Equations (1) through (7). This makes it possible to calculate an equivalent stress–time series from a time-dependent stress tensor.

Equivalent stress–time series can easily be calculated for proportional stress–time series by using this procedure. However, as sufficiently known from the literature, problems arise when applying it to nonproportional stresses, which can be easily illustrated by two examples [11,36,37], shown in Figure 5:

- Example 1, Figure 5, left side: A dominant shear stress amplitude is superimposed with a high-frequency normal stress of small amplitude (Figure 5a). The corresponding principal stresses and course of the first invariant that determines the later determined sign are shown in Figure 5b. By looking at both the coordinate and principal stresses, one would assume that the small normal stress amplitude would have a rather minor impact on the fatigue damage. However, the normal stress causes the sign to alternate with the same frequency. This leads to cycles in the signed equivalent stresses that are not physically motivated (see jumps in equivalent stresses in Figure 5c). These would lead to higher damage sums than expected. This behavior is independent of the chosen strength hypotheses.
- Example 2, Figure 5, right side: A normal stress amplitude is superimposed with a constant shear stress. Although the influence of the mean shear stress would be expected to be small, the resulting amplitudes of the equivalent stresses are large and, again, would lead to a significant amount of calculated damage.

It must be stated that the time series of signed equivalent stresses may be applied to proportional loading. The transfer to nonproportional loading, even if there are no formal hurdles, is associated with acceptance of the consideration of physically unreasonable cycles. The reliability of such an approach must therefore be limited [37].

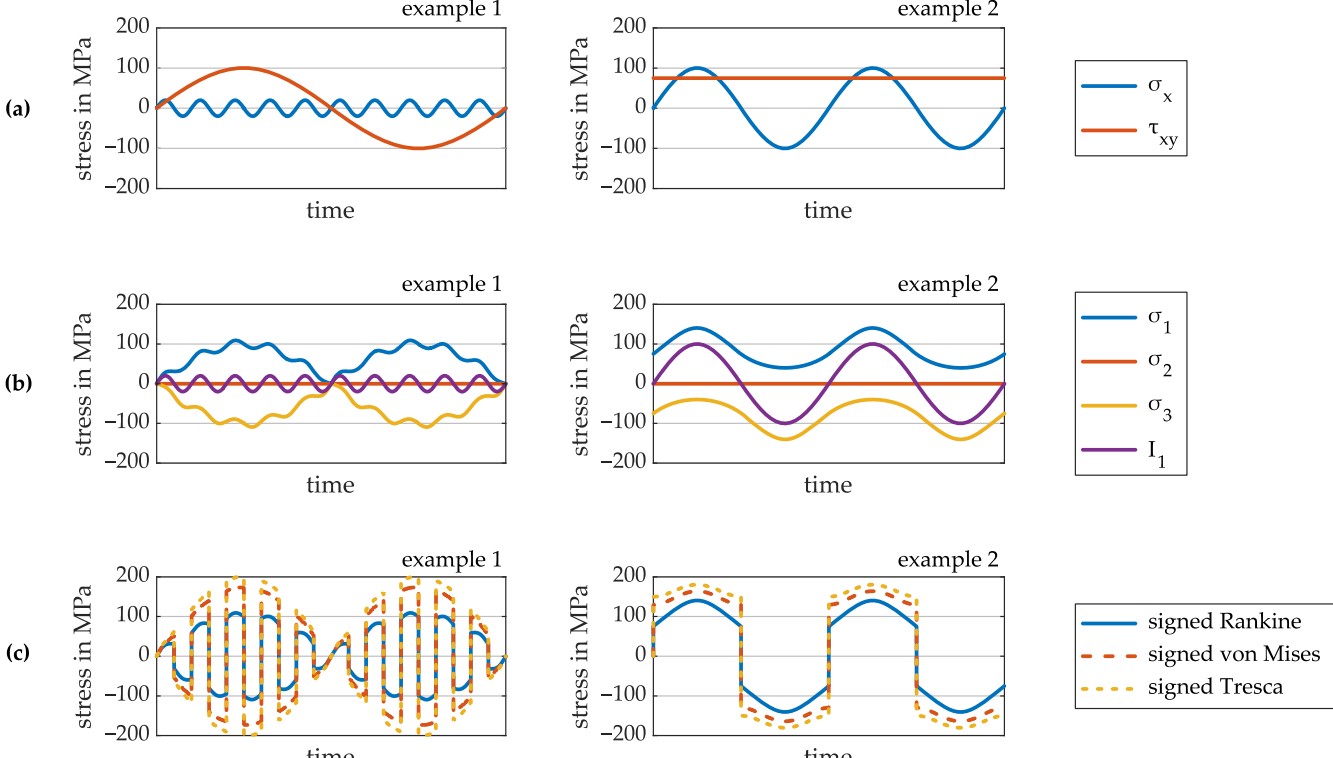

**Figure 5.** Two examples for combinations of normal and shear stresses that lead to (**a**) nonproportional loading and the resulting (**b**) principal stresses and first invariant and (**c**) signed equivalent stresses.

### 2.3. Scaled Normal Stresses in Critical Plane Approaches

Fatigue failure originates from and cracks grow along specific (critical) planes that are loaded by either dominant normal or shear stress depending on the material and loading conditions [13,38]. Aside from energy-based and empirically developed models, critical plane approaches are therefore commonly used to predict fatigue failure under nonproportional loading. In addition to the fatigue life in numbers of cycles, these models also predict the orientation of crack or failure planes [38].

Numerous critical plane approaches, which are not listed here, have been developed so far. Detailed reviews on this topic can be found [13,39–42].

One approach, originally suggested by Gaier and Dannbauer, is the concept of scaled normal stresses [9,10] that determine the critical plane based only on its normal stresses. The damaging influence of shear stresses is considered by scaling the normal stresses depending on the overall stress state and the material ductility. Another approach that uses the same basic concept was later suggested by Riess et al. [11,12]. Both approaches shall be examined in more detail below.

For understanding the following paragraphs, the plane projection for the normal stress shall be introduced at this point: The normal stress $\sigma_{[i]}$ for any plane i with normal vector $\mathbf{n}_{[i]}$ is given by Equation (13).

$$\sigma_{[i]} = \boldsymbol{\sigma}\mathbf{n}_{[i]} \circ \mathbf{n}_{[i]} \tag{13}$$

Here, $\circ$ is the operator for the scalar product and $\boldsymbol{\sigma}$ is the stress tensor.

$$\boldsymbol{\sigma} = \begin{pmatrix} \sigma_{xx} & \tau_{xy} & \tau_{xz} \\ \tau_{xy} & \sigma_{yy} & \tau_{yx} \\ \tau_{xz} & \tau_{yx} & \sigma_{zz} \end{pmatrix} \tag{14}$$

2.3.1. Scaled Normal Stresses According to Gaier and Dannbauer

Gaier and Dannbauer [9,10] propose a critical plane approach that uses the normal stress acting on a plane. These normal stresses (within the different planes) are scaled depending on two influences:

- The overall stress state that is independent of the individual planes is characterized by the ratio of principal stresses V:

$$
V = \begin{cases} \frac{\sigma_3}{\sigma_1} \text{ for } |\sigma_1| \geq |\sigma_3| \\[2ex] \frac{\sigma_1}{\sigma_3} \text{ for } |\sigma_3| \geq |\sigma_1| \end{cases}
\tag{15}
$$

The ratio V leads to values between −1 and +1. Three unique values result for specific stress states:

- Dominant shear stress (torsion): V = −1;
- Dominant tension/compression: V = 0;
- Hydrostatic stress: V = 1.

- The material ductility is represented by the value $f_{\tau/\sigma}$ according to Equation (4) for values between $f_{\tau/\sigma} = 0.5$ and 1.

The proposed calculation procedure can be structured as follows:

1. Calculating and sorting principal stresses $\sigma_1(t)$, $\sigma_2(t)$, and $\sigma_3(t)$ for each time step t of stress–time series $\boldsymbol{\sigma}(t)$;
2. Calculation of the ratio V(t) for each time step, Equation (15);
3. Calculation of scaling factor f(t):

$$
f(t) = 1 + \left(1 - \frac{1}{f_{\tau/\sigma}}\right) \cdot V(t)
\tag{16}
$$

4. Scaling the stress tensor for each time step:

$$
\overline{\boldsymbol{\sigma}}(t) = f(t) \cdot \boldsymbol{\sigma}(t)
\tag{17}
$$

5. The calculation of the time series for normal stress $\sigma_{eq,[i]}(t)$ for m individual planes i = 1, ... , m by projecting the scaled stress tensor:

$$
\sigma_{eq,[i]}(t) = \overline{\boldsymbol{\sigma}}(t)\mathbf{n}_{[i]} \circ \mathbf{n}_{[i]}
\tag{18}
$$

6. The equivalent stress–time sequence $\sigma_{eq,[i]}(t)$ can then be fed into a cycle counting method, e.g., rainflow counting. The resulting amplitudes are compared to an S–N curve, optionally after mean stress consideration, so that a damage calculation can be performed. In this way, a fatigue life $N_{[i]}$ is calculated for each considered plane. The smallest resulting fatigue life identifies the critical plane and represents the relevant fatigue life for the component.

As pointed out by Gaier and Dannbauer, steps 4 and 5 can also be executed in reverse order without changing the result; i.e., the scaling factor could also be applied to the normal stress after the projection to the plane. However, the procedure shown above is more computationally efficient since the operation of scaling needs to be conducted just once and not for each candidate plane.

While Lee, Barkey, and Kang [43] explicitly recommended the scaled normal stresses for "fatigue damage assessments of structures under variable amplitude, multiaxial loading conditions", McKelvey and Lee attribute the method with low accuracy in the case of shear failure mode [44].

2.3.2. Scaled Normal Stresses According to Riess et al. for Plane Stress States

Riess et al. develop an approach that is, like the one of Gaier and Dannbauer, based on the normal stresses in individual planes that are scaled depending on the overall stress state and the material ductility [11,12].

Riess et al. restrict their approach to plane stress states ($\sigma_{xx} \neq 0$, $\sigma_{yy} \neq 0$, $\tau_{xy} \neq 0$, i.e., $\sigma_1 \neq 0$, $\sigma_2 \neq 0$) and use the degree of multiaxiality $h_{biax}$ instead of the ratio V to characterize the entire stress tensor.

$$h_{biax} = \begin{cases} \frac{\sigma_2}{\sigma_1} \text{ for } |\sigma_1| \geq |\sigma_2| \\[2mm] \frac{\sigma_1}{\sigma_2} \text{ for } |\sigma_2| \geq |\sigma_1| \end{cases} \tag{19}$$

$h_{biax}$ leads to values between $-1$ and $+1$. Three unique values result for specific stress states:

- Dominant shear stress (torsion): $h_{biax} = -1$;
- Dominant tension/compression: $h_{biax} = 0$;
- Biaxial stress state: $h_{biax} = 1$.

The material ductility is also considered through the value $f_{\tau/\sigma}$ according to Equation (4). Here, however, the range for lies between 0.58 and 1.

Compared to Section 2.3.1, the calculation procedure of Riess et al. therefore differs concerning the following steps:

1.  Calculating and sorting principal stresses $\sigma_1(t)$ and $\sigma_2(t)$ for each time step t of stress–time series $\sigma(t)$;
2.  Calculation of degree of multiaxiality $h_{biax}(t)$ for each time step, Equation (19);
3.  Calculation of scaling factor f(t):

$$f(t) = \left[ \frac{1}{1 - \left( \frac{1-f_{\tau/\sigma}}{1-\frac{1}{\sqrt{3}}} \right) \cdot \left( 1 - \frac{1}{\sqrt{1+h_{biax}^2(t)-h_{biax}(t)}} \right)} \right] \tag{20}$$

4.  Scaling of stress tensor for each time step:

$$\overline{\sigma}(t) = f(t)\sigma(t) \tag{21}$$

5.  Calculation of time series for normal stress $\sigma_{eq,[i]}(t)$ for individual planes $i = 1, \dots, m$ by projecting the scaled stress tensor:

$$\sigma_{eq,[i]}(t) = \overline{\sigma}(t)\mathbf{n}_{[i]} \circ \mathbf{n}_{[i]} \tag{22}$$

Even though the approach according to Riess et al. and that of Gaier and Dannbauer are comparable in steps 1 to 5, Riess' proposal differs with respect to the further processing of the equivalent stress within the individual planes. It is suggested that they be used in combination with either a stress-based concept (6.a) or a strain-based concept (6.b):

6.a   *Stress-based concept:* Similarly to step 6 in Section 2.3.1, Riess et al. additionally show how changes in the influence of stress gradients throughout the load series that are due to nonproportional loading can be taken into account. In addition, a suggestion is made for dealing with the decrease in fatigue life due to nonproportional hardening, which is explained in more detail in Section 2.4.

6.b   *Strain-based concept:* The equivalent stress–time sequence $\sigma_{eq,[i]}(t)$ is fed into a uniaxial notch root approximation (Neuber's rule [45]) and a rainflow counting method [46] so that elastic–plastic stress–strain hysteresis can be simulated and closed hysteresis loops can be detected as load cycles. Damage parameter values according to Smith,

Watson, and Topper (SWT [47]) are then calculated for each hysteresis loop, and Miner's rule can be applied against an SWT–Wöhler curve.

Riess et al. account for nonproportional hardening by implementing the proposal of Socie and Marquis [13] of modifying the cyclic strength coefficient of the material by using a nonproportionality factor [48] (see Section 2.4).

Riess et al. point out that the term $\frac{1}{\sqrt{1+h_{biax}^2(t)-h_{biax}(t)}}$ within the scaling factor in Equation (20) describes the ratio between the equivalent stresses of Rankine $\sigma_{eq,R}$ and von Mises $\sigma_{eq,M}$ (for plane stress states, $\sigma_3 = 0$):

$$\frac{\sigma_{eq,R}}{\sigma_{eq,M}} = \begin{cases} \dfrac{\sigma_1}{\sqrt{\sigma_1^2+\sigma_2^2-\sigma_1\sigma_2}} = \dfrac{1}{\sqrt{\frac{\sigma_1^2}{\sigma_1^2}+\frac{\sigma_2^2}{\sigma_1^2}-\frac{\sigma_2}{\sigma_1}}} = \dfrac{1}{\sqrt{1+h_{biax}^2-h_{biax}}} \text{for} |\sigma_1| \geq |\sigma_2| \\[4mm] \dfrac{\sigma_2}{\sqrt{\sigma_1^2+\sigma_2^2-\sigma_1\sigma_2}} = \dfrac{1}{\sqrt{\frac{\sigma_1^2}{\sigma_2^2}+\frac{\sigma_2^2}{\sigma_2^2}-\frac{\sigma_1}{\sigma_2}}} = \dfrac{1}{\sqrt{1+h_{biax}^2-h_{biax}}} \text{for} |\sigma_2| \geq |\sigma_1| \end{cases} \quad (23)$$

### 2.4. Effects under Nonproportional Loading

From the results of cyclic testing of either material specimens (unnotched) and components or component-like specimens (notched) under both proportional and nonproportional loading, two phenomena are known to occur that shall be discussed in more detail below, namely (1) nonproportional (strain) hardening of the material (Section 2.4.1) and (2) shifts in fatigue life to either higher or lower number of cycles compared to proportional loadings (Section 2.4.2).

2.4.1. Nonproportional Hardening

Nonproportional (strain) hardening (also out-of-phase hardening) is an effect that occurs to a greater or lesser extent in certain materials. It leads to hardening of the material under nonproportional loading in comparison to proportional loading with the same strain amplitudes. The effect is attributed to the fact that different slip systems are addressed by the rotating principal axis system as it passes through the load sequence leading to cross slip [49]. Meanwhile, a relationship is suggested between a material's capacity for nonproportional hardening and the stacking fault energy. The higher the stacking fault energy, the lower the assumed nonproportional hardening, e.g., as for many aluminum materials [13].

The amount of nonproportional hardening can experimentally be determined in strain-controlled fatigue tests on thin-walled tubular specimens under combined axial and torsional load. While the axial load leads to normal stresses and strains, the torsional load leads to almost pure shear stresses and strains with negligible shear stress/strain gradients.

To describe the nonproportional hardening of a material, Socie and Marquis introduced the nonproportional hardening coefficient $\alpha$ [13].

$$\alpha = \frac{\varepsilon_{a,p}(\text{nonproportional loading})}{\varepsilon_{a,p}(\text{proportional loading})} - 1 \quad (24)$$

This describes the difference between the plastic strain amplitude at nonproportional and proportional stress at the same strain amplitude. The plastic strain amplitude in the numerator of Equation (24), on the one hand, can be determined by tests with uniaxial loading or by tests with proportionally combined axial and torsional loading. The plastic strain amplitude in the denominator on the other hand is determined in tests with nonproportionally combined axial and torsional loading. In order to obtain a single plastic strain amplitude from a nonproportional loading, an equivalent strain amplitude must be calculated during the test evaluation. The procedure for this is described by Savaidis [50] or Shamsaei et al. [51].

In cases where a material law such as that according to Ramberg and Osgood [52], Equation (25), is used to describe the material, $\alpha$ may also be interpreted as the difference between the strain hardening coefficients for nonproportional loading with 90° phase shift

$K'_{np}$ and uniaxial or proportional loading $K'$ (Equation (26) and Figure 6a). This requires that the same cyclic strain hardening exponent is used for both types of loading $n'_{np} = n'$. Otherwise, $\alpha$ would be dependent on the strain amplitude ($\alpha \neq$ const.).

$$\varepsilon_a = \frac{\sigma_a}{E} + \left(\frac{\sigma_a}{K'}\right)^{\frac{1}{n'}} \tag{25}$$

$$\alpha = \frac{K'_{np}}{K'} - 1 \tag{26}$$

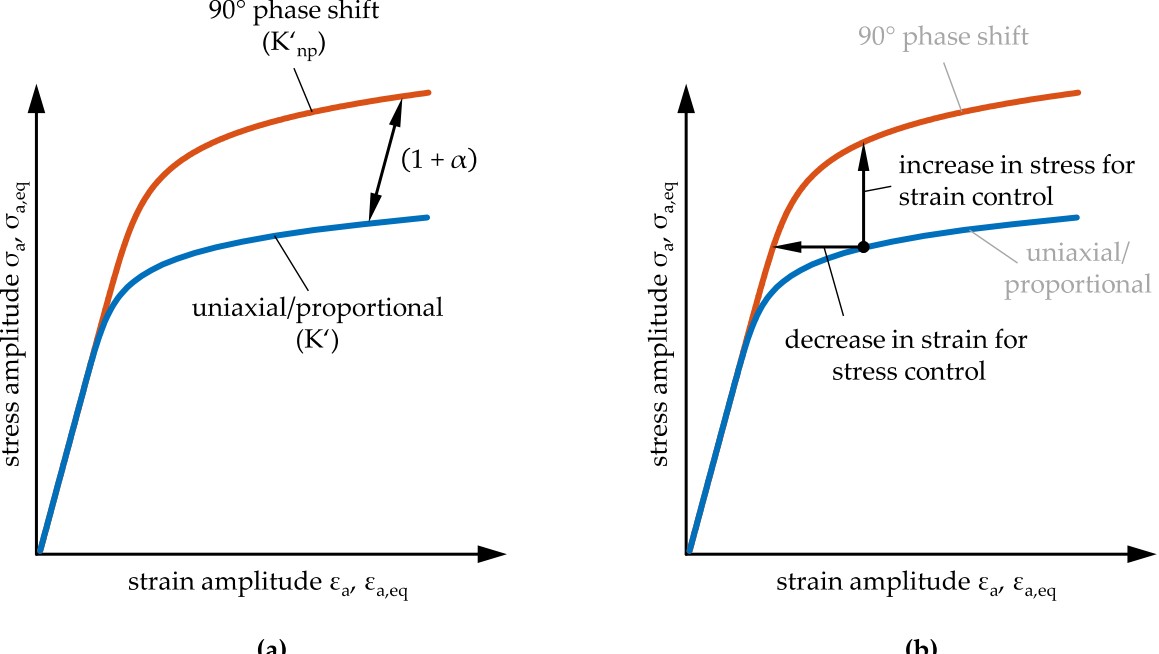

**Figure 6.** Nonproportional hardening expressed as (**a**) the difference between cyclic stress–strain curves for uniaxial or proportional and nonproportional loading with 90° phase shift and (**b**) the effect on the resulting stress–strain amplitudes under nonproportional loading depending on the type of control.

Since the highest amount of nonproportional hardening is usually expected for 90° phase shift between normal and shear strain [13], tests for determining $K'_{np}$ are usually performed for this phase shift. $\alpha$ shows values between 0 and 1 [13].

Since $\alpha$ is costly to determine, Borodii and Shukaev attempt to approximate the quantity based on static hardening coefficients [53]. Shamsaei and Fatemi later find strong correlations between nonproportional hardening on the one hand and cyclic and monotonic uniaxial deformation behavior on the other hand [51]. A developed approximation method yields promising results.

Sonsino points out that, on the one hand, in unnotched specimens under strain control, nonproportional hardening leads to a reduction in fatigue life compared to proportional loading. Stress-controlled tests on unnotched specimens, on the other hand, would result in an increase in fatigue life [54]. This may be explained by the increase or decrease in local stress and strain amplitudes depending on the type of control (stress or strain control) (Figure 6b).

To describe the nonproportional hardening within the cyclic stress–strain curve for different phase shifts, an additional nonproportionality factor F is necessary:

$$\varepsilon_a = \frac{\sigma_a}{E} + \left(\frac{\sigma_a}{K' \cdot (1 + \alpha \cdot F)}\right)^{\frac{1}{n'}} \tag{27}$$

F is a measure for the nonproportionality of an applied load. Multiple definitions for F have been suggested (see, e.g., compilations in [40,48]), which shall not be discussed here. However, for most definitions of F, proportional loading results in F = 0 and maximum nonproportionality (e.g., 90° phase shift) results in F = 1.

2.4.2. Shift in Fatigue Life during Nonproportional Loading

Next to the results that can be found for unnotched specimens, Sonsino compiles experimental results [14,54] for notched specimens under nonproportional loading. Sonsino concludes that notched specimens experience shifts in fatigue life to either higher number of cycles (increase in fatigue life) or lower number of cycles (decrease in fatigue life) compared to proportional loadings with the same amplitudes of the individual stress components. Some materials also show no significant difference in fatigue life for proportional and nonproportional loading and are attributed neutral behavior [14].

Sonsino also finds that the tendency toward increasing or decreasing fatigue life is not connected to whether the tests are conducted under load or displacement control as was found for unnotched specimens [54] (compare with Section 2.4.1). The findings of Sonsino may be explained by the fact that the local stresses and strain within the notch are connected to controlled loads such as in Neuber control (Figure 7).

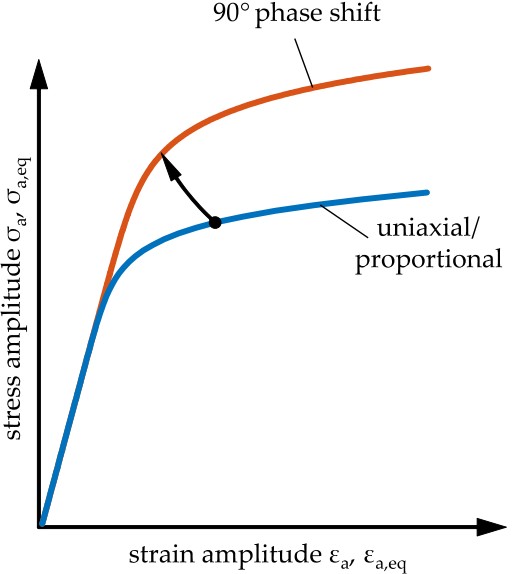

**Figure 7.** Expected shift of local stress and strain amplitude during nonproportional hardening for notched components.

Sonsino finds that a decrease in fatigue life can be found mostly for rather ductile materials, neutral behavior for medium ductile materials, and an increase in fatigue life for rather brittle metals, although exceptions to these observations also exist [14]. By looking at the examples compiled by Sonsino, it can be found that the decreases or increases in fatigue life appear almost as parallel shifts of the S–N curves for proportional and nonproportional loading. However, exceptions can also be found for this in Sonsino's survey.

Sonsino connects the described observations to the blockade of activated dislocations and, thus, to nonproportional cyclic hardening (compare with Section 2.4.1) [14].

The effect of shifts in fatigue life under nonproportional loading may also be connected to the same mechanisms described in Section 2.4.1, and the authors see one inconsistency in the explanation solely for nonproportional hardening: a significant difference between the local stress–strain states between proportional and nonproportional loading should only result if the loading is in a range that leads to noteworthy plastic strain components (at least in the case of proportional loading). At lower load levels, by contrast, where only very limited plastic strain occurs even in the notch root, hardly any influence should be

discernible. With these arguments, it would be expected that different slopes of the S–N curves are evident with proportional and nonproportional loading. However, the test results compiled by Sonsino show, in many cases, an approximately parallel shift of the S–N curves and, thus, a significant influence of nonproportionality on the fatigue life range close to the endurance limit. Therefore, the question is raised whether for nonproportional loading, in addition to the effect of different combinations of local stresses and strains, an influence on the component strength or its fatigue life has to be considered. In the opinion of the authors, this question has not yet been conclusively answered.

Nevertheless, the effect of a shift in the fatigue life of notched geometries under nonproportional loading has been known for some time; i.e., Gaier et al. [55] developed a simple correction to account for this phenomenon in analytical fatigue assessments by shifting the S–N curve of a material for nonproportional loading (Figure 8).

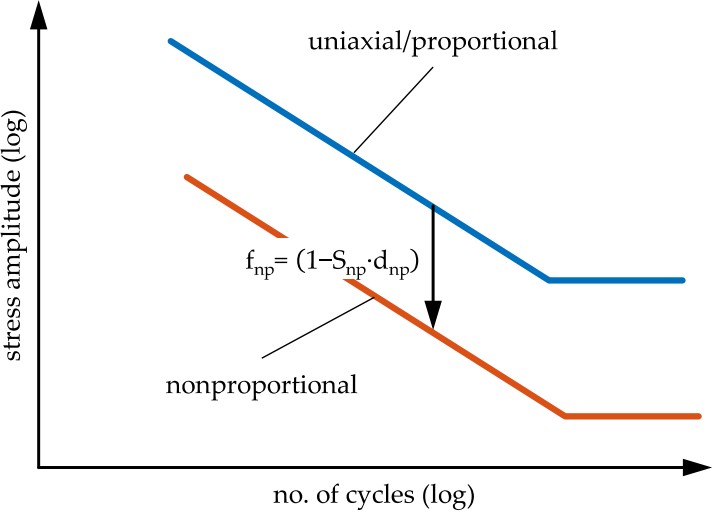

**Figure 8.** A shift of local S–N curve due to nonproportional loading as suggested by Gaier et al. [55].

The factor for shifting the S–N curve is thereby defined in the direction of stress as a factor $f_{np}$ between the endurance limits of proportional loading $\sigma_{E,np}$ and nonproportional loading $\sigma_{E,prop}$:

$$f_{np} = \frac{\sigma_{E,np}}{\sigma_{E,prop}} = \left(1 - S_{np} \cdot d_{np}\right) \tag{28}$$

Here, $d_{np}$ is a measure for the nonproportionality of the load, similar in function to the quantity F in Equation (27); Snp is an adjustment factor to account for the shifting of fatigue life or strength under nonproportional loading.

## 3. Comparison of the Two Approaches for Scaled Normal Stresses

The two approaches for scaled normal stresses have obvious similarities and differences. These approaches and their implications will be compared below.

In both approaches, the normal stress in the candidate plane is used as the critical stress component. It is scaled depending on the overall stress state present at each point in time as well as the material ductility. The scaled normal stress calculated in this way is the equivalent stress for the subsequent fatigue life calculation in each candidate plane.

The approach of Gaier and Dannbauer can be used for triaxial stress states, i.e., also in cases where critical points located inside the component or at component surfaces are subjected to pressure or friction forces. The approach of Riess et al. is permissible for the case of plane stress states only, which is present at unloaded component surfaces and, therefore, sufficient for many applications.

The scaling factors in the two approaches differ not only in terms of their characteristic value for the stress state but also in terms of the permissible range of material ductility. While Gaier and Dannbauer [9,10] assign the value $f_{\tau/\sigma}$ =0.5 to a ductile material,

Riess et al. [11,12] use the value $f_{\tau/\sigma}$ =0.58. In both cases, brittle material is assigned the value $f_{\tau/\sigma}$ =1. In the latter case, normal stresses in the plane are not scaled at all. For $f_{\tau/\sigma}$ =1, the results are as follows:

$$f_{\text{Gaier}}(t) = 1 + \left(1 - \frac{1}{f_{\tau/\sigma}}\right) \cdot V(t) = 1 + \left(1 - \frac{1}{1}\right) \cdot V(t) = 1 \qquad (29)$$

$$f_{\text{Riess}}(t) = \frac{1}{1 - \left(\frac{1-f_{\tau/\sigma}}{1-\frac{1}{\sqrt{3}}}\right) \cdot \dots} = \frac{1}{1 - \left(\frac{1-1}{1-\frac{1}{\sqrt{3}}}\right) \cdot \dots} = 1 \qquad (30)$$

Therefore, the equivalent stress is identical to the normal stress. The used strength hypothesis is the one of Rankine (Figures 1 and 2b).

In the case of fully ductile material ($f_{\tau/\sigma}$ =0.5), the following scaling factor results for Gaier and Dannbauer:

$$f_{\text{Gaier}}(t) = 1 + \left(1 - \frac{1}{f_{\tau/\sigma}}\right) \cdot V(t) = 1 + \left(1 - \frac{1}{0.5}\right) \cdot V(t)$$

$$= 1 - V(t) = \begin{cases} 1 - \frac{\sigma_3(t)}{\sigma_1(t)} = \frac{\sigma_1(t) - \sigma_3(t)}{\sigma_1(t)} \text{ for } |\sigma_1| \geq |\sigma_3| \\[2mm] 1 - \frac{\sigma_1(t)}{\sigma_3(t)} = \frac{\sigma_3(t) - \sigma_1(t)}{\sigma_3(t)} \text{ for } |\sigma_3| \geq |\sigma_1| \end{cases} \qquad (31)$$

The last transformation step shows that the scaling factor is the ratio between the equivalent stresses according to Tresca $\sigma_{\text{eq,T}}$ and Rankine, comparing Equations (1) and (2). It could therefore also be written as follows:

$$f_{\text{Gaier}}(t) = \frac{\sigma_{\text{eq,T}}}{\sigma_{\text{eq,R}}} \qquad (32)$$

As has been shown before, Equation (23), the scaling factor of Riess et al., contains the ratio between the equivalent stresses of Rankine and von Mises. For fully ductile material ($f_{\tau/\sigma} = \frac{1}{\sqrt{3}} = 0.58$), the following scaling factor results:

$$f_{\text{Riess}}(t) = \frac{1}{1 - \left(\frac{1-f_{\tau/\sigma}}{1-\frac{1}{\sqrt{3}}}\right) \cdot \left(1 - \frac{\sigma_{\text{eq,R}}}{\sigma_{\text{eq,M}}}\right)} = \frac{1}{1 - \left(\frac{1-\frac{1}{\sqrt{3}}}{1-\frac{1}{\sqrt{3}}}\right) \cdot \left(1 - \frac{\sigma_{\text{eq,R}}}{\sigma_{\text{eq,M}}}\right)} = \frac{1}{1 - \left(1 - \frac{\sigma_{\text{eq,R}}}{\sigma_{\text{eq,M}}}\right)} = \frac{\sigma_{\text{eq,M}}}{\sigma_{\text{eq,R}}} \qquad (33)$$

For both approaches, in the case of proportional loading, the critical planes are identical to the planes in which the principal stresses with the largest absolute values are present. The associated normal stresses, again, are identical to the resulting equivalent stresses according to Rankine. By scaling these normal stresses in the critical plane with the factors in Equations (32) and (33), identical results are obtained in the case of a fully ductile material according to the approach of Gaier and Dannbauer and for those obtained with the signed equivalent stress according to Tresca. Likewise, the results according to Riess et al. are identical to the signed equivalent stress according to von Mises. It should be noted that the identity of the results is given only if the sign of the signed equivalent stresses is determined by the principal stress with the largest absolute value, Equation (9), or in case of plane stress states (compare with Section 2.2).

The critical plane hypotheses are validated not only by the fact that the fatigue life predicted with these realistically reflects the experimental findings but also that the match in the orientations of calculated and measured planes of initiated fatigue cracks. When using the scaled normal stress hypothesis, there is a setback in this regard. In the case of shear-dominated crack initiation, it may be possible to estimate the fatigue life with acceptable accuracy, but not the crack orientation. For a potential user, the advantage of ease of use compensates for this drawback.

By varying the values of $f_{\tau/\sigma}$, the strength hypotheses used in the two approaches can be represented in the form of principal stress diagrams for plane stress (Figure 9).

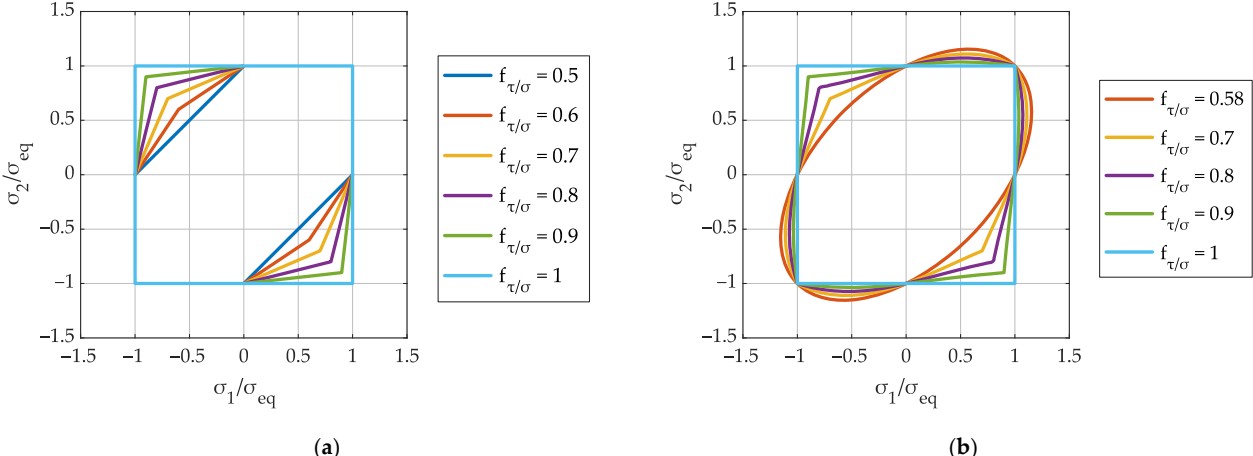

**Figure 9.** Strength hypothesis according to (**a**) Gaier and Dannbauer and (**b**) Riess et al. for plane stress states ($\sigma_3 = 0$).

At first glance, the approaches of Gaier–Dannbauer and Riess et al. seem to provide the same results as the mixed hypotheses of El-Magd and Lüpfert–Spies, respectively, for proportional stresses. On closer examination, however, this is only confirmed for Gaier–Dannbauer and El-Magd. Between Riess et al. and Lüpfert–Spies, there are small but recognizable differences. These differences originate from different mixture terms that control the influence between von Mises and Rankine (compare Equations (7), (8) and (33)).

Identical results, however, can be confirmed for the Gaier–Dannbauer and El-Magd pairing for triaxial stress states. The approach according to Riess et al., however, is only permissible for plane stress states.

Below, the scaled normal stress approaches are applied to the nonproportional stress series from the two examples in Section 2.2. The course of (scaled) normal stress of three exemplary candidate planes ($\varphi = 0$, $\frac{\pi}{8}$, $\frac{\pi}{4}$) is shown in Figure 10b–d. Since both examples deal with plane stress states, only planes perpendicular to the surface need to be considered as candidate planes, and one angle ($\varphi$) suffices for the plane definition. $\varphi = 0$ corresponds to the coordinate system of the stresses in Figure 5. Figure 10a also shows the resulting scaling factors for the stress tensor for both approaches.

In this case, the values of $f_{\tau/\sigma} = 0.5$ and $0.58$ are chosen for fully ductile material for the approaches of Gaier–Dannbauer and Riess et al. The stress series that would result for $f_{\tau/\sigma} = 1$ (fully brittle material) is also provided in Figure 10 since it is identical to the unscaled normal stresses in the planes for both approaches.

Figure 10 shows that neither of the two problems that occur in the calculation of signed equivalent stresses is relevant for the scaled normal stress approaches: in neither case do jumps in the stress signal lead to unphysical cycles, nor do mean or static stresses of single stress components have a negative effect on individual signals due to excessively large amplitudes.

It shall be noted that the value $f_{\tau/\sigma} = 0.58$ might also be used for fully ductile materials in the Gaier–Dannbauer approach. In this case, the resulting values for the scaling factor and the scaled normal stresses for both approaches in Figure 10 would be almost identical.

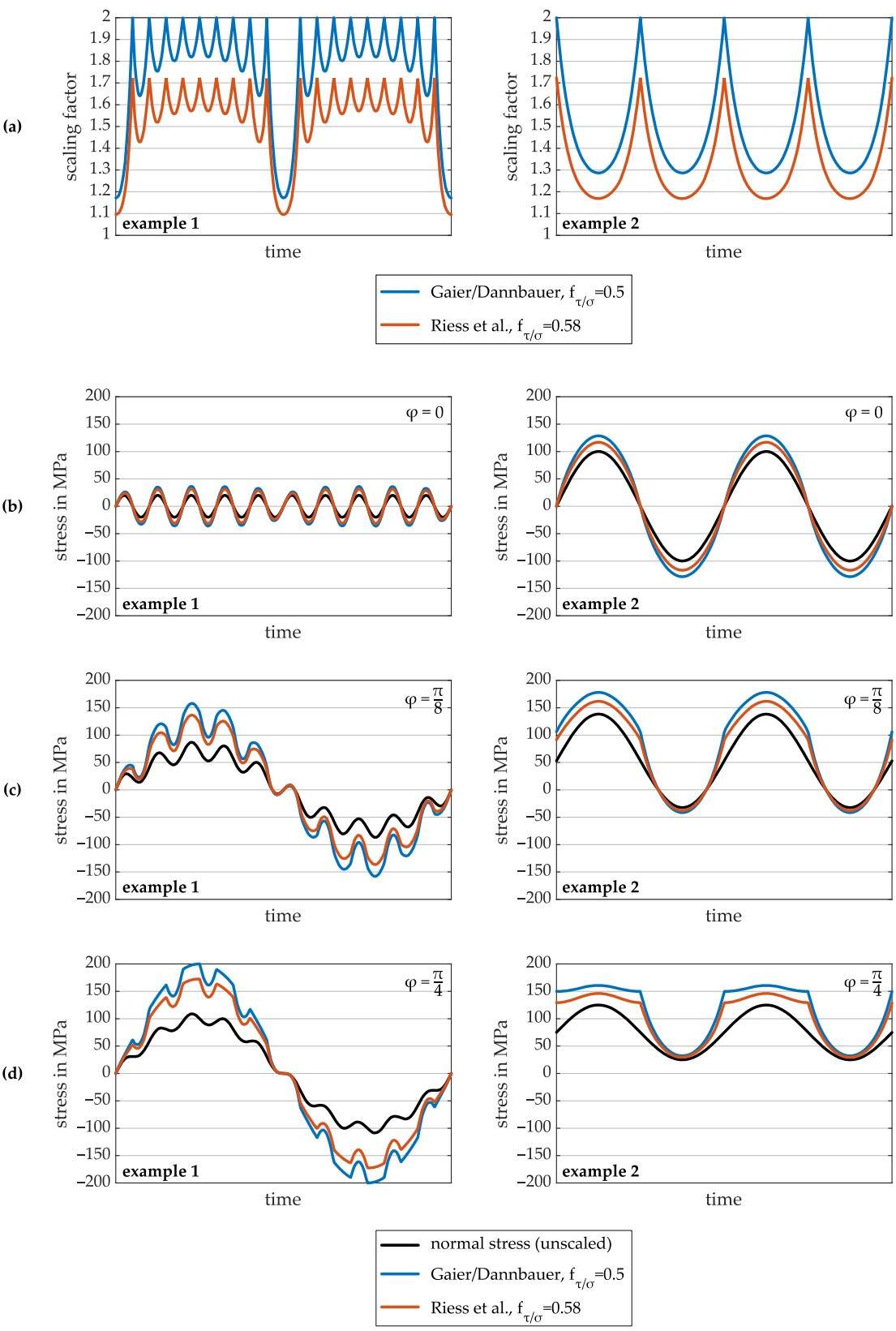

**Figure 10.** Time series for (**a**) scaling factors and (**b**–**d**) scaled normal stresses in different planes for the approach according to Gaier–Dannbauer and Riess et al. for the examples 1 and 2 of nonproportional stress series in Figure 5.

## 4. Generalization of the Approach for Scaled Normal Stresses

From the elaborations in the previous sections, it can be recognized that the following points are characteristic for the two known approaches of the scaled normal stresses:

1.  The normal stress is used as the relevant stress component within the candidate planes. It therefore determines the sign of the equivalent stress according to this approach.
2.  The ductility of the material, represented by the quantity $f_{\tau/\sigma}$, and the share of shear stresses in the global stress state are taken into account using a scaling factor for the normal stress within the candidate planes.

As has been shown by the considerations in Section 3, in principle, superpositions from two different strength hypotheses are used within the scaling factors. Gaier and Dannbauer use the ratio between Tresca and Rankine hypothesis, Equation (32). Riess et al. use the ratio between von Mises and Rankine hypothesis, Equation (33). This is (unintentionally) disguised by the use of the quantities $V$ and $h_{biax}$, both of which characterize the global stress state.

The influence of the ratios between either Tresca or von Mises, respectively, and Rankine depends on the ductility of the material as described by $f_{\tau/\sigma}$. The approach of Gaier and Dannbauer uses the same hypothesis as El-Magd in the case of proportional loading and is therefore fully compatible. Compared to the hypothesis of Lüpfert and Spies, the approach of Riess et al. leads to similar but nonidentical results for plane stress.

The procedure of the two approaches can be generalized as follows: The equivalent stress $\sigma_{eq,X}$ of an arbitrary strength hypothesis X is put into relation with the equivalent stress according to Rankine. The scaled normal stresses in a candidate plane result:

$$\sigma_{eq,[i]}(t) = \left( \frac{\sigma_{eq,X}}{\sigma_{eq,R}} \cdot \boldsymbol{\sigma}(t) \right) \mathbf{n}_{[i]} \circ \mathbf{n}_{[i]} \tag{34}$$

Since the relevant stress component in the planes is the normal stress, the reference value in the denominator of the scaling factor needs to be the equivalent stress according to Rankine.

Depending on which strength hypothesis is used in the numerator of the scaling factor, different compatibilities can be established with the hypotheses accepted for proportional stresses. In addition, the material ductility can be taken into account. Various strength hypotheses can be selected, including at least the ones mentioned in Section 2.1:

1.  *Rankine:* Insertion for $\sigma_{eq,X}$ in Equation (34) would be possible. However, the scaling factor would always be 1. For fully brittle material, therefore, no scaling factor would be necessary.
2.  *von Mises or Tresca:* Possible for fully ductile materials.
3.  *El-Magd:* Provides a superposition of Tresca and Rankine depending on the ductility $f_{\tau/\sigma}$ and leads, with a modified formula, to identical results to the approach according to Gaier and Dannbauer.
4.  *Lüpfert and Spies:* Provides a superposition of von Mises and Rankine depending on the ductility $f_{\tau/\sigma}$.

Variant 4 gives very similar but not identical results to the approach of Riess et al. for plane stress states but is also valid for triaxial stress states and fully compatible with the approach known for proportional loading.

## 5. Exemplary Application to Ductile Cast Iron

The use of scaled normal stresses is to be tested by means of an example. For this purpose, the variant presented in Section 4 is used (item 4 in Section 4), namely Equation (34) in combination with the strength hypothesis of Lüpfert and Spies. This is applied to experimental results from notched specimens under bending and torsional load. The material of the specimens is ductile cast iron (nodular graphite cast iron) EN-GJS-500-14 (material No. 5.3109) according to EN 1563 [56] and is expected to have a ratio of fatigue

strength for shear and normal stresses $f_{\tau/\sigma}$ between 0.5 and 1 (e.g., $f_{\tau/\sigma}$ = 0.65, refer to Table 1). To characterize the amount of nonproportional hardening of the material, the deformation behavior of unnotched tubular specimens is also investigated.

As described above, scaled normal stresses could be used within either a stress- or a strain-based approach. In the strain-based concepts, it is not possible to use Wöhler curves determined on components (in this case, notched specimens) as a basis for calculation. However, these S–N curves determined on components already take into account significant influences of the component itself, which have to be estimated in the strain-based concept by means of additional dependencies. As examples, support effects due to the notch geometry or different S–N curve inclinations for material and component can be mentioned here [57]. In the following considerations, however, the focus shall be on the difference between proportional and nonproportional loading, which is why the application of scaled normal stresses is shown using a stress-based approach. An S–N curve determined under proportional loading is used as the basis for the calculation since it is assumed to give matching properties for the notched specimen.

### 5.1. Experimental Results

Multiaxial fatigue tests were conducted on specimens of EN-GJS-500-14. Below, a short overview of the test conditions and results is given, which is necessary for the application of scaled normal stresses. A detailed explanation of the tests is provided in Appendix A.

The monotonic properties of the material were determined in tensile tests according to ISO 6892-1 [58] on proportional tensile specimens and according to [59] with a diameter in the test range of 8 mm. The results are listed in Table 2, where the determined elastic modulus is also provided.

**Table 2.** Monotonic properties of EN-GJS-500-14.

| Property | Value |
|---|---|
| tensile strength | 550 MPa |
| 0.2% yield strength | 444 MPa |
| elongation at rupture | 17% |
| elastic modulus | 165,000 MPa |

Different types of fatigue tests were performed on the described material. At first, load-controlled tests with constant load amplitudes were conducted on notched cylindrical specimens (Figure 11a). The specimens have a notch radius of 2 mm. This leads to elastic stress concentration factors of $K_{t,b} \approx 1.83$ for bending and $K_{t,t} \approx 1.42$ for torsion. Loading with the bending moment $M_b$ leads to local linear elastic normal stresses of $\sigma_x = 1.0998 \times 10^{-3} \frac{MPa}{Nmm} \cdot M_b$ and $\sigma_y = 2.2972 \times 10^{-4} \frac{MPa}{Nmm} \cdot M_b$ within the notch. The coordinate system in the notch is defined in Figure 11a. The axes x and y are parallel to the specimen surface. Loading with the torsional moment $M_t$ leads to local shear stresses $\tau_{xy} = 4.4987 \times 10^{-4} \frac{MPa}{Nmm} \cdot M_t$.

The specimens were subjected to:

1. Pure bending;
2. Pure torsion;
3. Proportional loading with combined bending and torsion with amplitude ratio $\tau_{xy,a}/\sigma_{x,a} = 0.75$;
4. Nonproportional loading with combined bending and torsion while the torsional loading shows a phase shift of 90° in reference to bending, amplitude ratio $\tau_{xy,a}/\sigma_{x,a} = 0.75$.

All tests were conducted with fully reversed loading, meaning a stress ratio of $R = \frac{\sigma_{min}}{\sigma_{max}} = -1$.

The tests were conducted until the specimen ruptured into two halves; meanwhile, the notch was monitored using several cameras that recorded pictures of the specimen surface every 10 to 1000 load cycles, depending on the expected number of cycles. Thus, after the

experiment, it was possible to evaluate the number of cycles for crack initiation of surface cracks of about 0.5 to 1.5 mm in length.

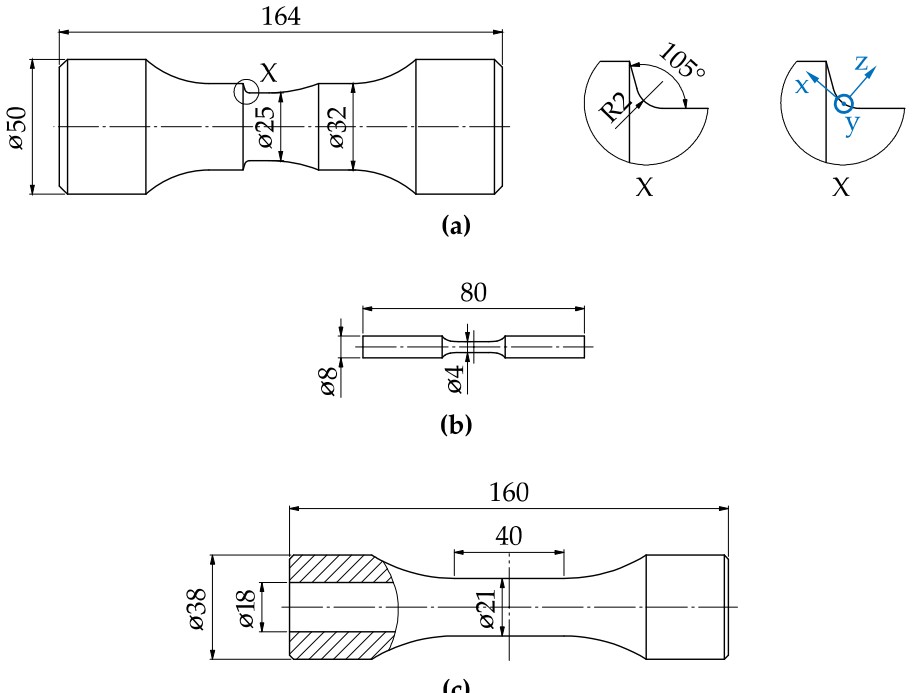

**Figure 11.** Specimen geometries used for fatigue tests with important dimensions: (**a**) notched specimen for load-controlled tests, (**b**) unnotched axial specimen for strain-controlled tests, (**c**) tubular unnotched specimen for normal and shear strain-controlled tests.

The numbers of cycles determined in this way are plotted as S–N curves in Figure 12 using the local linear elastic stress amplitudes that result from the loads of bending and/or torsion. For the tests with bending components, only the amplitude of the stress component $\sigma_x$ is shown, and for the S–N curve with pure torsion, the shear stress $\tau_{xy}$ is displayed. The exponent k and the fatigue strength at $10^5$ cycles $\sigma_{1E5}$ of the Basquin equation [60]

$$N = 10^5 \cdot \left( \frac{\sigma_a}{\sigma_{1E5}} \right)^{-k} \leftrightarrow \sigma_a = \sigma_{1E5} \cdot \left( \frac{N}{10^5} \right)^{-\frac{1}{k}} \tag{35}$$

that result from least square regression can be found in Appendix A.

By comparing the S–N curves for proportional and nonproportional loading that are based on the same ratio between bending and torsion amplitudes, a decrease in fatigue life for the nonproportional results can be found.

Apart from the tests of notched specimens, the nonproportional hardening of the material was also characterized. Therefore, first, strain-controlled tests were performed on unnotched axial specimens (Figure 11b) to obtain the cyclic stress–strain curve according to Ramberg and Osgood that describes the uniaxial deformation behavior. The resulting cyclic stress–strain curve is plotted in Figure 13. The cyclic strain hardening exponent n' and the cyclic strain hardening exponent K' are listed in Table 3. Figure 13 also contains the monotonic stress–strain curve from the beginning of one of the individual tests. It can be seen that the material shows cyclic hardening.

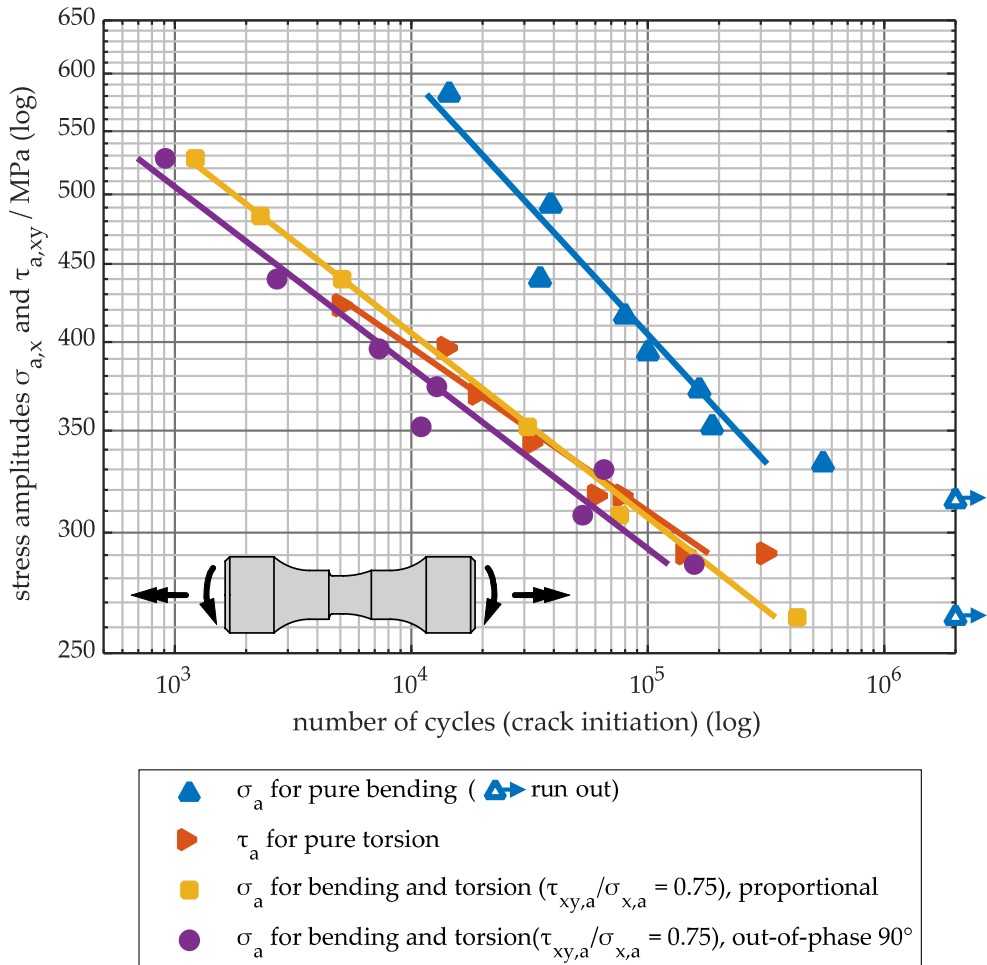

**Figure 12.** S–N curves for load-controlled tests on notched specimens, with amplitudes of local linear elastic stress components.

Second, tests on unnotched tubular specimens (Figure 11c) were carried out under axial tension and torsion. The two loads were applied with a phase shift of 90° between each other. For the evaluation of the test with nonproportional loading, refer to Section 2.4. The values for resulting properties $K'_{np}$ and $n'_{np}$ can be found in Table 3. In order to exclude the influence of specimen shape on the uniaxial and multiaxial nonproportional results, a few tests with uniaxial tension and tests with proportional axial and torsional loading were also carried out on the tubular specimens. All results are plotted beside the uniaxial results from the small axial specimens (Figure 13). The uniaxial and proportional test results on the tubular specimens are in good agreement with the uniaxial results from the small axial specimens.

**Table 3.** Cyclic properties according to Ramberg and Osgood for uniaxial and nonproportional (90° phase shift) loading.

| Loading | Property | Value |
|---|---|---|
| uniaxial/ proportional | E | 165,000 MPa |
| | K′ | 830 MPa |
| | n′ | 0.0801 |
| nonproportional (90° phase shift) | E | 165,000 MPa |
| | $K'_{np}$ | 1153 MPa |
| | $n'_{np} = n'$ | 0.0801 |

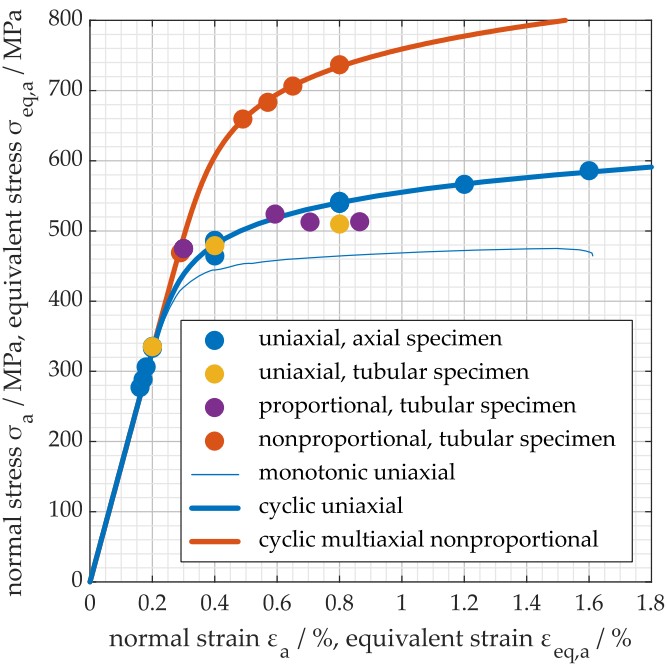

**Figure 13.** Comparison of results for uniaxial monotonic and cyclic deformation behavior with uniaxial, proportional, and nonproportional loading.

In Figure 14, the normal stress–normal strain hysteresis and the shear stress–shear strain hysteresis from tests with strain amplitudes of $\varepsilon_a = 0.40\%$ and $\gamma_a = 0.76\%$ for both proportional and nonproportional loading are shown. The additional nonproportional hardening can clearly be recognized due to the increase in the stress amplitude for nonproportional loading. In addition, the typical rounding of the hysteresis loops due to the phase shift can be seen.

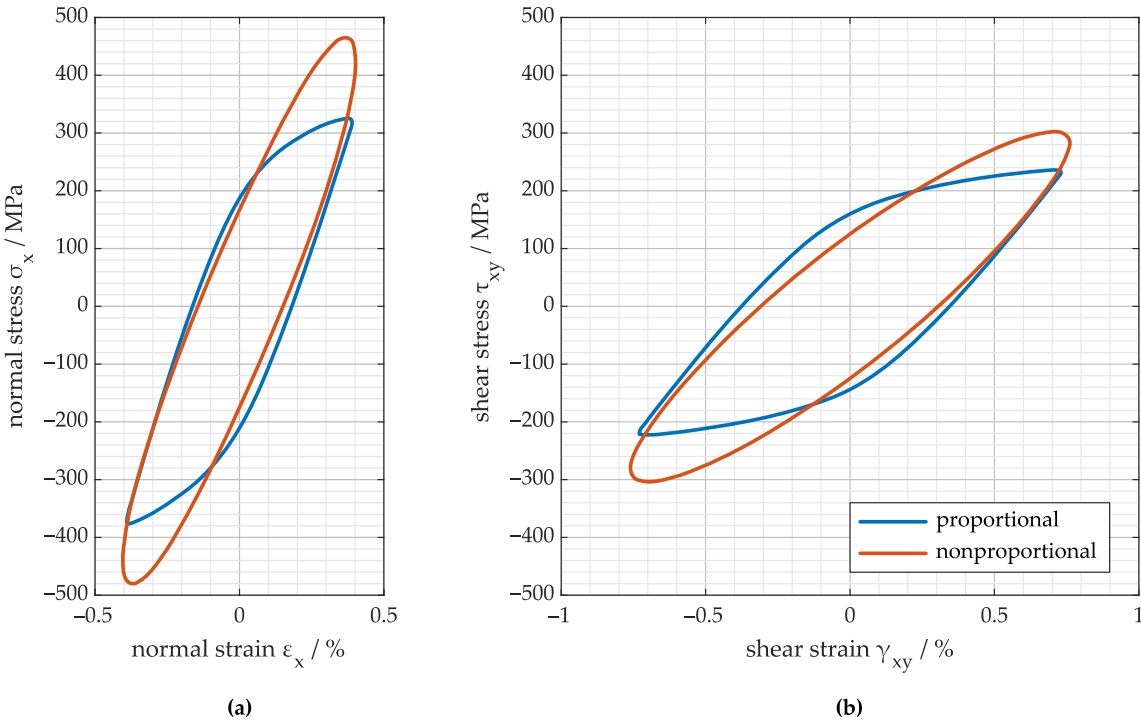

**(a)**                                                                      **(b)**

**Figure 14.** Stress–strain hysteresis for proportional loading and nonproportional loading with phase shift of 90° at the same normal and shear strain amplitudes: (**a**) normal stresses and strains and (**b**) shear stresses and strains.

The nonproportionality parameter $\alpha$ is determined according to Equation (36) (see Section 2.4.1):

$$\alpha = \frac{K'_{np}}{K'} - 1 = \frac{1153 \text{ MPa}}{830 \text{ MPa}} - 1 = 0.39 \qquad (36)$$

### 5.2. Application of Scaled Normal Stresses

The results from the notched specimens shown before are evaluated using the approach of scaled normal stresses according to Equation (34) in combination with the strength hypothesis of Lüpfert and Spies (item 4 in Section 4).

For the application, a ratio of fatigue strength for shear and normal stresses, $f_{\tau/\sigma}$, needs to be specified. For the material under investigation, different slopes result for the two S–N curves for pure bending and pure torsional loading, which reflects the usual behavior. It is therefore not surprising that there is no value for $f_{\tau/\sigma}$ that describes all fatigue life ranges equally well. For shorter fatigue lives, the value tends to 0.6, and for longer fatigue lives it is around 0.75. In the following, the average value of $f_{\tau/\sigma} = 0.7$ is used.

For all S–N curves from Figure 12, the equivalent stress amplitudes according to the approach of scaled normal stresses are calculated and plotted over the experimental fatigue lives for each test result (Figure 15a). It can be seen that the results for pure bending, pure torsion, and their proportional combinations fall in one scatter band that may be described by one S–N curve (black curve in Figure 15a). This proportional S–N curve shows an exponent of $k = 7.7$ and a strength at $10^5$ load cycles of $\sigma_{1E5} = 394$ MPa.

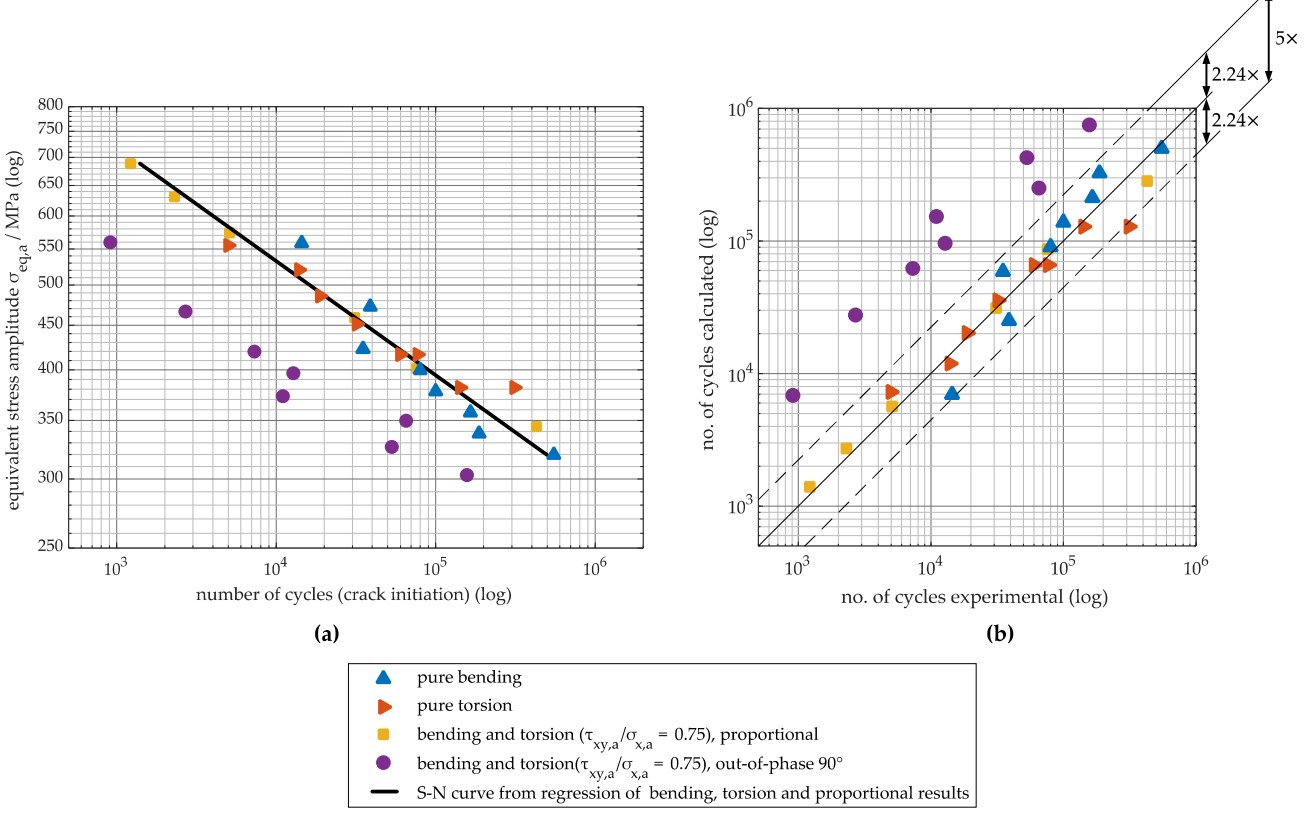

**Figure 15.** Test results from Figure 12 evaluated with scaled normal stresses as (**a**) equivalent stress amplitudes according to scaled normal stresses and (**b**) number of cycles calculated based on the proportional S–N curve over an experimentally determined number of cycles.

This S–N curve is used to calculate a fatigue life for each test result, which makes it possible to plot the calculated vs. experimentally determined number of cycles (Figure 15b). Calculated results above the angle bisector tend to be unsafe because the calculated lifetime

is larger than the experimentally determined lifetime. Figure 15b also contains the scatter band for factor 5 around the angle bisector ($\pm$ factor $\sqrt{5} = 2.24$ to the angle bisector). It can be seen that the results for pure bending, pure torsion, and proportional loading almost all lie within the factor 5 scatter band.

It can also be seen from Figure 15 that the decrease in fatigue life for nonproportional hardening cannot be reproduced by the scaled normal stresses without further modification. This is not surprising since none of the properties connected with nonproportional hardening were considered in the calculation. The calculation results for nonproportional hardening are overestimated by approximately one decade.

By comparing Figures 12 and 15b, one also finds that the distances between the results for proportional and nonproportional loading are much larger if the experiments are plotted as scaled normal stresses (Figure 15b) in comparison to the results for the amplitude of the normal stress component $\sigma_{a,x}$ (Figure 12). This is due to the formation of the equivalent stress of the scaled normal stress, which considers that, in contrast to proportional loading, the amplitudes of the individual stress components do not occur at the same time for nonproportional loading. The equivalent stress amplitude of the scaled normal stress is therefore smaller for nonproportional loading than for proportional loading.

*5.3. Corrections for Nonproportional Loading*

In this section, possible corrections for nonproportional loading are explored.

As a first approach, an attempt was made to use the experimentally proven nonproportional hardening of the material for the correction of the nonproportional load case. The effect of nonproportional hardening, on the one hand, is connected to plastic deformations. The scaled normal stresses, on the other hand, are evaluated on a stress-based approach with linear elastically assumed material behavior for this contribution. In order to be able to describe the difference in the plastic deformation behavior for proportional and nonproportional loading, a notch root approximation according to Neuber [45] is used. This shall be described in more detail:

1. To approximate the local elastic–plastic stresses and strains from the linear elastically determined ones, a notch root approximation is used, as is often done when applying strain-based calculation concepts (cf. [38,61,62]).

   In this case, the notch root approximation of Neuber [45] that postulates the equality of the products of stress and strain for both linear elastic ($\sigma_a$ and $\varepsilon_a$) and elastic–plastic ($\sigma_a^{e-p}$ and $\varepsilon_a^{e-p}$) material behavior (Equation (37) or Equation (38)) is applied.

$$\sigma_a \cdot \varepsilon_a = \sigma_a^{e-p} \cdot \varepsilon_a^{e-p} \tag{37}$$

$$\frac{\sigma_a^2}{E} = \sigma_a^{e-p} \cdot \varepsilon_a^{e-p} \tag{38}$$

   Here, E is the elastic modulus.

   By combining Equation (38) with the cyclic stress–strain curves either according to Equation (25) for proportional loading or according to Equation (27) for nonproportional loading, pairs of $\sigma_a^{e-p}$ and $\varepsilon_a^{e-p}$ can be determined for a linear elastic stress amplitude $\sigma_a$ for both load cases.

   For the proportional load case, the elastic–plastic material behavior is given by the Ramberg–Osgood equation with the properties E = 165,000 MPa, K' = 830 MPa, and n' = 0.0801, whereas for the nonproportional case with 90° phase shift, the properties E = 165,000 MPa, $K'_{np}$ = 1153 MPa, and $n'_{np}$ = 0.0801 are used (cf. Table 3).

   For practical application and implementation of notch root approximation, refer to the work of Burghardt et al. [63].

2. The approximated elastic–plastic stress amplitudes for proportional and nonproportional loading are used to calculate the quotient between both (Figure 16a):

$$n_{np} = \frac{\sigma_a^{e-p} \text{ (nonproportional)}}{\sigma_a^{e-p} \text{ (proportional)}} \tag{39}$$

The value of this quotient $n_{np}$ depends on the load level (see Figure 16b). Figure 16b shows the factor $n_{np}$ obtained for the material at hand for various linear elastic stress amplitudes. Additionally, the range of values of the stress amplitudes of the experiments between 300 and 560 MPa is shown.

3. The quotient $n_{np}$ is used as a correction factor to consider the nonproportional hardening in combination with the linear elastically determined scaled normal stresses. It can be used to either increase the equivalent stress amplitudes for nonproportional loading (Figure 17a) or to actually decrease the strength of the proportional S–N curve (Figure 17b).

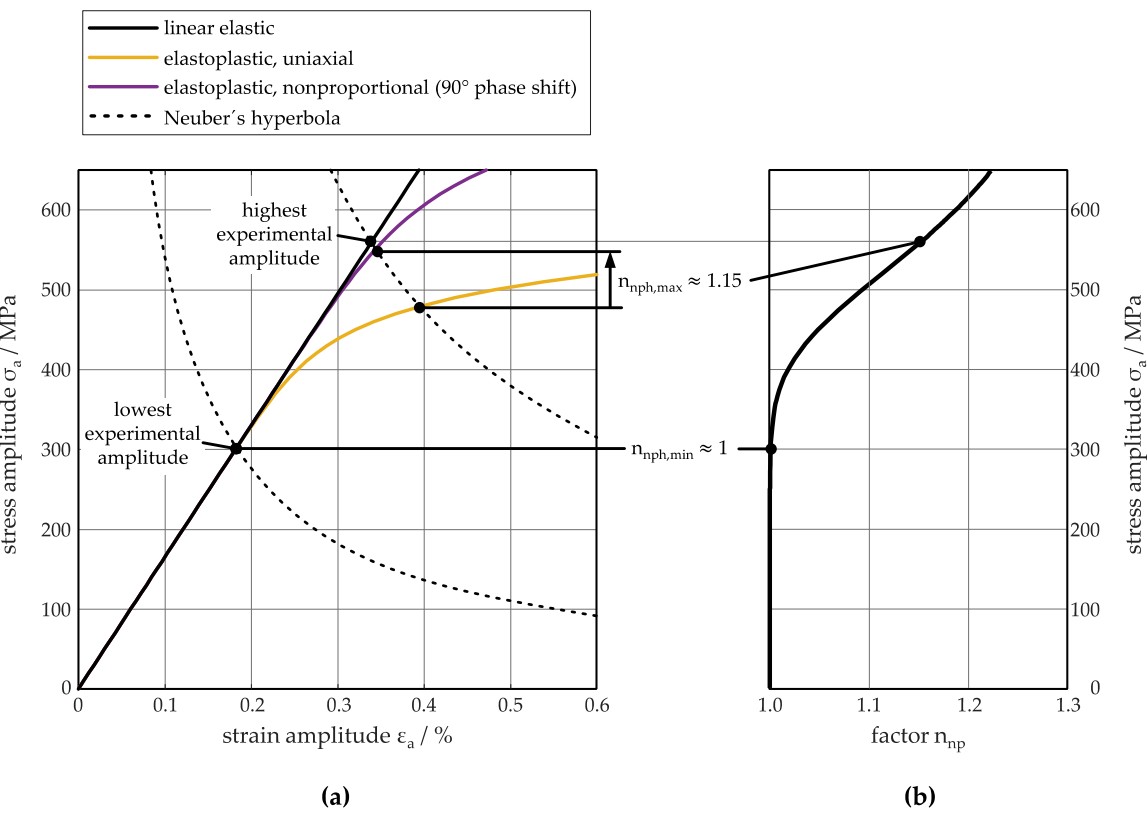

**Figure 16.** Determination of load–height-dependent correction factor $n_{np}$ for nonproportional hardening: (**a**) calculation of correction factor through notch root approximations according to Neuber and (**b**) values for $n_{np}$ for linear elastic stress amplitudes.

Figure 17 shows that the chosen approach that only considers nonproportional hardening is not sufficient to correct for nonproportional loading. The correction in the region of the experimental results with the highest loads of 560 MPa leads to $n_{np} = 1.15$ and seems to tend toward the right direction; however, it must be stated that the correction is still unsatisfactory and still produces results on the unsafe side. Furthermore, and not surprisingly, remembering the reasoning from Section 2.4.1, when considering only nonproportional hardening, appreciable effects, if there are any, will only be observed in the case of high load levels.

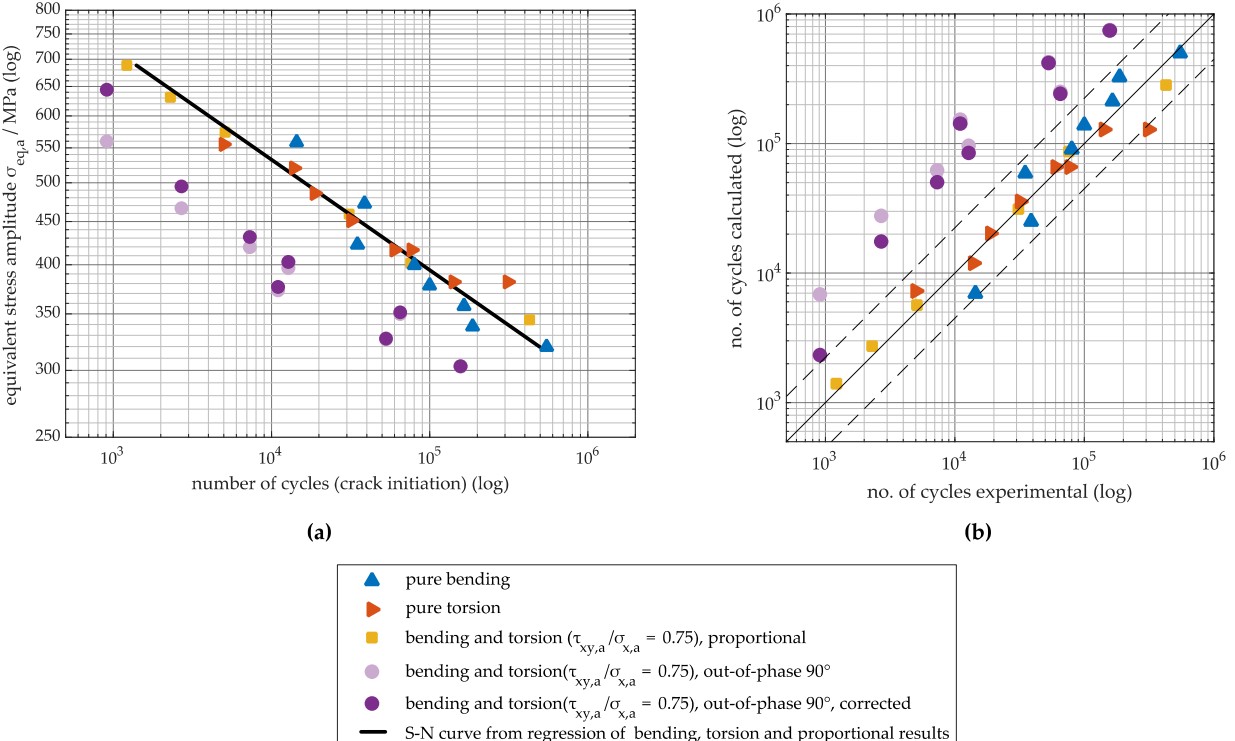

**Figure 17.** Test results from Figure 12 evaluated with scaled normal stresses and results for nonproportional loading corrected with factor for *nonproportional hardening $n_{np}$*: (**a**) equivalent stress amplitudes according to scaled normal stresses and (**b**) number of cycles calculated based on the proportional S–N curve over an experimentally determined number of cycles.

Therefore, a second attempt to correct for nonproportional loading was made: Gaier et al. [55] proposed shifting the S–N curve for proportional loading using Equation (28). This technique can also be applied in the present exemplary case. For this purpose, it was not necessary to specify a certain formulation for the nonproportionality measure $d_{NP}$; instead, the product $S_{np} \cdot d_{np}$ may be empirically adjusted: the best fit between nonproportional and the results for proportional loading as well as for pure bending and pure torsion was found for $S_{np} \cdot d_{np} = 0.25$. This leads to a factor $f_{np} = 0.75$, meaning a reduction in fatigue strength of the S–N curve for proportional loading to 75%.

$$f_{np} = \left(1 - S_{np} \cdot d_{np}\right) = \left(1 - 0.25 \cdot 1\right) = 0.75 \tag{40}$$

For later practical application, a formulation for $d_{np}$ must obviously be selected, leading to the following formulation of the S–N curve based on Equation (35):

$$\sigma_a = f_{np} \cdot \sigma_{1E5} \cdot \left(\frac{N}{10^5}\right)^{-\frac{1}{k}} = \left(1 - S_{np} \cdot d_{np}\right) \cdot \sigma_{1E5} \cdot \left(\frac{N}{10^5}\right)^{-\frac{1}{k}} \tag{41}$$

However, the value of the sensitivity $S_{np}$ would also have to be adapted to this choice. The two values must always be seen in combination. However, clarification of this subaspect is not the focus of this contribution.

Figure 18 results if this factor $f_{np} = 0.75$ is used to either increase the equivalent stress amplitudes for nonproportional loading (Figure 18a) or to actually decrease the strength of the proportional S–N curve (Figure 18b).

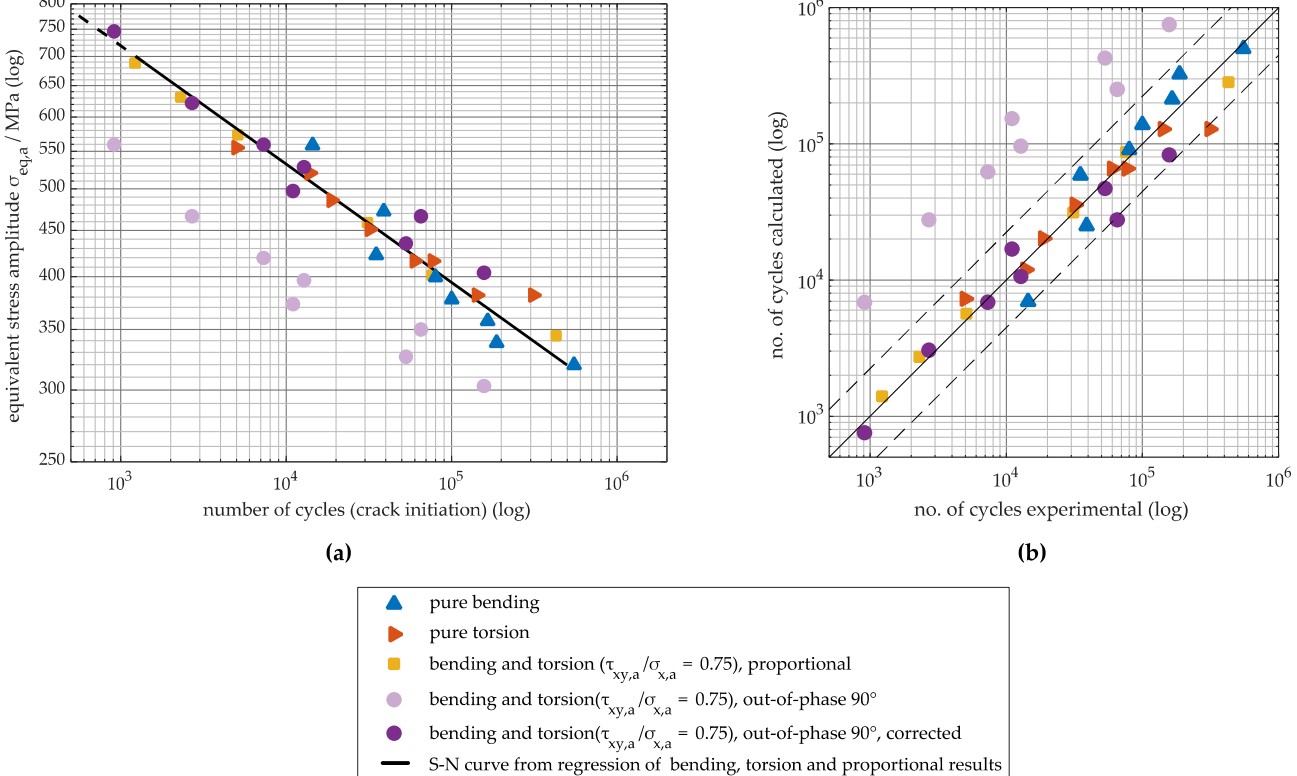

**(a)**　　　　　　　　　　　　　　**(b)**

| | |
|---|---|
| ▲ (blue) | pure bending |
| ▶ (orange) | pure torsion |
| ■ (yellow) | bending and torsion ($\tau_{xy,a}/\sigma_{x,a}$ = 0.75), proportional |
| ● (light purple) | bending and torsion($\tau_{xy,a}/\sigma_{x,a}$ = 0.75), out-of-phase 90° |
| ● (dark purple) | bending and torsion($\tau_{xy,a}/\sigma_{x,a}$ = 0.75), out-of-phase 90°, corrected |
| —— | S-N curve from regression of bending, torsion and proportional results |

**Figure 18.** Test results from Figure 12 evaluated with scaled normal stresses and results for non-proportional loading corrected with *factor according to Gaier et al.*: (**a**) equivalent stress amplitudes according to scaled normal stresses and (**b**) number of cycles calculated based on the proportional S–N curve over an experimentally determined number of cycles.

It can be seen that after correction with the factor $f_{np}$, the results for the nonproportional load fit well into the scatter band of the other results. Thus, almost all calculated values for fatigue life lie within the scatter band for factor 5.

It can be stated that the correction according to Gaier et al. leads to satisfactory correction of the nonproportional results. Of course, this is also due to the fact that the sensitivity factor was empirically determined in this case.

## 6. Conclusions and Outlook

In this study, the two variants of the critical plane approach of the scaled normal stresses of Gaier–Dannbauer and Riess et al. were closely examined. The following has been found:

1. The method of Gaier–Dannbauer is compatible with the strength hypothesis of El-Magd for the proportional case if the latter is used as a signed equivalent stress.
2. For the biaxial proportional case, the method of Riess et al. yields similar but not identical results to the strength hypothesis of Lüpfert–Spies if the latter is used as a signed equivalent stress.
3. The effects typically encountered when using a signed equivalent stress with nonproportional loadings, such as excessive amplitudes and unphysical sign changes, do not occur with the scaled normal stresses.
4. The principle of scaled normal stresses can be generalized in the form of Equation (34) in such a way that various strength hypotheses can be used in the scaling factor. The material ductility can potentially be taken into account depending on the strength hypothesis used, as in the original proposals by Gaier–Dannbauer or Riess et al. By using the strength hypothesis of Lüpfert–Spies, the restriction of the approach of Riess et al. to biaxial stress states can be circumvented.

5. When applying the generalized variant described under item 4 with the strength hypothesis according to Lüpfert–Spies, the results from tests with constant amplitude loading on notched bending and torsion specimens of ductile cast iron EN-GJS-500-14 demonstrated a good description of the results for the proportional case. The results for pure bending, pure torsion, and proportional combinations of both fall into one scatter band.

6. Nonproportional effects cannot be accounted for by the scaled normal stress approach without modification. Two methods for correcting the nonproportional loading case have been investigated:

   a. The first accounts for the nonproportional hardening of the investigated material and uses notch root approximations for the estimation of the elastic–plastic stress amplitudes for both proportional and nonproportional loading. In principle, this approach shifts the experimental results in the right direction but is not sufficient for satisfactory correction. In addition, the consideration of nonproportional hardening only has an effect in the case of high load levels.

   b. The second approach uses a correction factor, empirically determined according to Gaier et al., that decreases the whole S–N curve for nonproportional loading. Using a factor of $f_{np} = 0.75$ to decrease the S–N curve leads to satisfactory results, whereby the nonproportional and proportional results lie within one common scatter band.

7. It has thus been exemplarily shown that the method of scaled normal stresses is a flexible tool available for the evaluation of multiaxial fatigue strength. This tool requires the input of only a few material properties. On the one hand, the material ductility expressed by the ratio of the fatigue strengths for normal and shear stress $f_{\tau/\sigma}$ and a uniaxial or proportional S–N curve are required. For both quantities, there are either blanket values available or possibilities for estimation from relatively easy to determine quantities. For $f_{\tau/\sigma}$, see material-group-specific values in Table 1. For the estimation of S–N curves, see Section 1. On the other hand, a quantity is needed for taking the nonproportional shift in fatigue life into account. For now, this quantity has been empirically determined, meaning fatigue tests on notched geometries under proportional and nonproportional loading were necessary.

In future studies, the following aspects should be addressed:

8. An accuracy evaluation should be carried out on the basis of as many test results as possible with constant and variable amplitude loadings. A comparison with other approaches for multiaxial fatigue must be made.

9. A suitable definition of a nonproportionality measure to account for different, also varying, phase shifts also needs to be chosen and combined with the proposed approaches. This is most important for a practical application of the approach and also very challenging since descriptions of experience with nonproportionality measures in combination with variable amplitudes are almost nonexistent throughout the literature.

10. For application in engineering practice, the accuracy should also be checked in cases where all material properties are estimated (cf. item 7). This is particularly necessary for the determination of safety factors for component approval.

11. The suitability of the factor $f_{np}$ proposed by Gaier et al., Equation (26), to account for nonproportional loading must be verified on the basis of more application cases and materials. The sensitivity factor $S_{np}$ needs to be determined for a wide variety of materials. For practical application, there should be further investigations of how to determine $S_{np}$ based on properties that are easier to determine than by using multiaxial fatigue tests on notched specimens.

12. Items 8 to 11 should also be conducted while using scaled normal stresses in combination with a strain-based approach. In particular, the consideration of nonproportional

hardening could be determined in a more sophisticated manner through material models, and therefore, such an approach could be more successful.

**Author Contributions:** Conceptualization, M.W. and A.L.; methodology, M.W., C.G. and M.V.; software, M.W. and A.L.; investigation, M.W., A.L., R.W., J.K. and C.F.; data curation, A.L.; writing—original draft preparation, M.W.; writing—review and editing, M.W., A.L., R.W., A.E., C.G., J.K., C.F. and M.V.; visualization, M.W. and R.W.; supervision, A.E. and M.V.; project administration, A.L., J.K., R.W. and C.F.; funding acquisition, A.E., M.V. and M.W. All authors have read and agreed to the published version of the manuscript.

**Funding:** This research was funded through two projects by the German *Federal Ministry for Economic Affairs and Climate Action* on the basis of a decision by the *German Bundestag*. These two projects of *Allianz Industrie Forschung (Aif),* having IGF Nos. 20613 and 21306, are professionally supervised by *Forschungskuratorium Maschienbau e.V. (FKM)*.

**Institutional Review Board Statement:** Not applicable.

**Informed Consent Statement:** Not applicable.

**Data Availability Statement:** The test results on EN-GJS-500-14 specimens presented in this study are available in Section 5.1 and Appendix A.

**Acknowledgments:** The authors express their gratitude to the foundry *SLR Giesserei St. Leon-Rot GmbH* in St. Leon-Rot, Germany, for providing the casting blanks made of EN-GJS-500-14 from which the fatigue specimens were manufactured and *ZF Friedrichshafen AG* in Passau, Germany, for financing the CNC machining of some of the fatigue specimens used. Furthermore, the authors thank *Allianz Industrie Forschung (Aif)* for the funding and *Forschungskuratorium Maschienbau e.V. (FKM)* for the professional supervision of the underlying research projects. In the end, the authors acknowledge the support of the Open Access Publishing Fund of Clausthal University of Technology.

**Conflicts of Interest:** The authors declare no conflict of interest.

## Appendix A. Test Conditions and Results for Fatigue Tests

In this appendix, detailed information is provided for the fatigue tests on EN-GJS-500-14 material that are described in the main section.

*Appendix A.1. Specimens*

The raw material for the specimens was available as raw cast cylinders of dimensions ∅ 60 mm × 250 mm (Figure A1). The specimens for the fatigue tests (Figure 11) were CNC-machined from these raw cylinders.

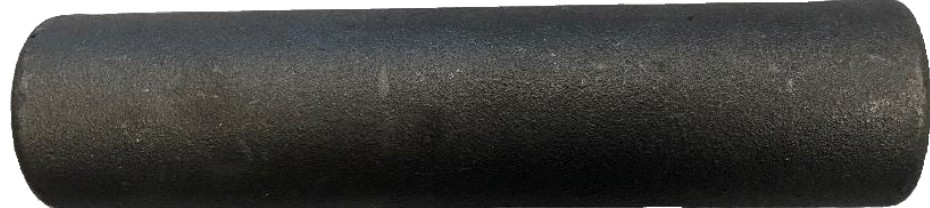

**Figure A1.** Raw cylinders cast from EN-GJS-500-14 material.

*Appendix A.2. Load-Controlled Fatigue Tests on Notched Specimens*

The tests were conducted on a servohydraulic testing machine that can apply both bending and torsion to the specimens (Figure A2).

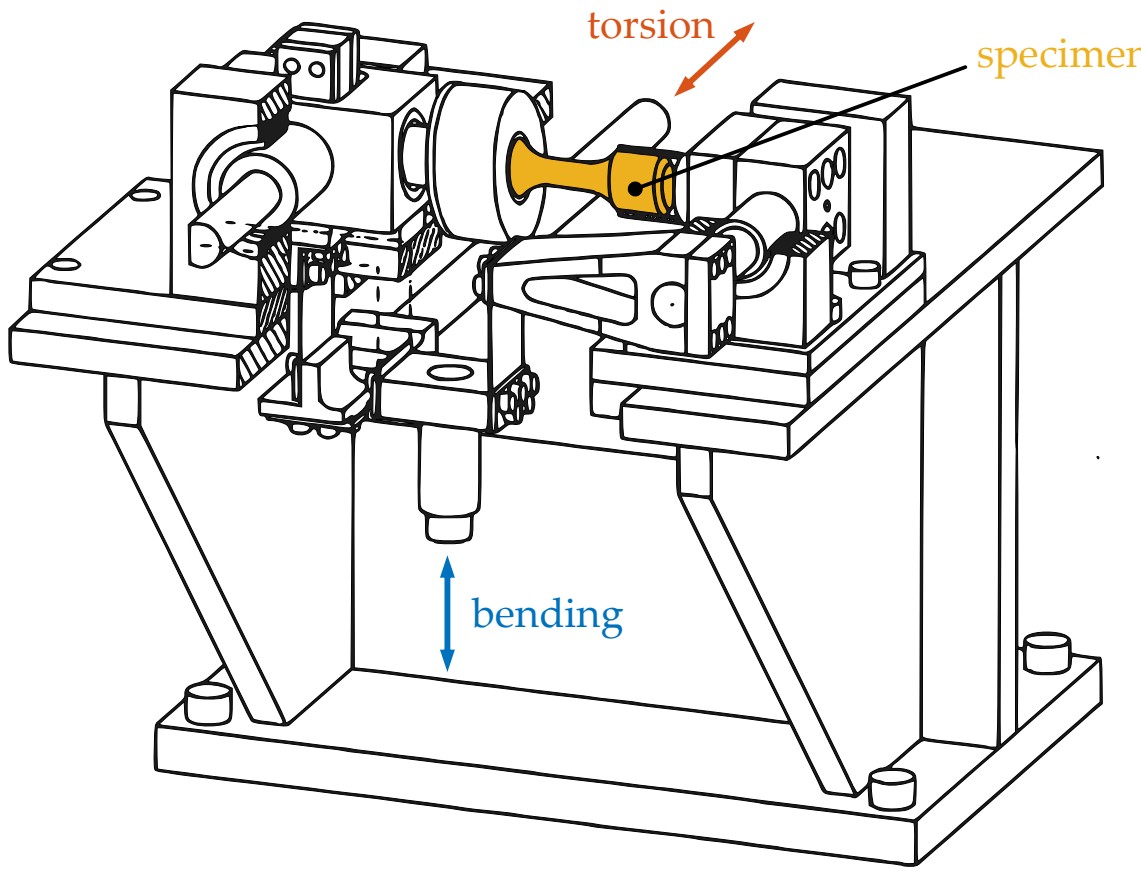

**Figure A2.** Scheme of testing machine used for load-controlled tests.

An MTS Flextest 100 controller controls the servohydraulic actuators using the software MTS 793 Controller software. All tests were conducted using sine waveforms for the load signals.

All the test results for the individual load-controlled fatigue tests at constant load amplitude are given in Tables A1–A4.

All numbers of cycles until crack initiation were determined after the completion of the individual tests by evaluating photos of the notch that were taken during the test every 10 to 1000 load cycles, depending on the expected number of cycles. The photos were taken by a camera system built in-house with eight cameras distributed around the circumference of the notch. This made it possible to reliably detect cracks in the range of 0.5 to 1.5 mm.

The test execution and evaluation were conducted according to DIN 50100 [64]. The test results using the local linear elastic stress amplitudes that result from the loads of bending and/or torsion are shown in Figure 12. For the tests with bending components, only the amplitude of the stress component $\sigma_x$ is shown, and for the S–N curve with pure torsion, the shear stress $\tau_{xy}$ is displayed. The exponent k and the fatigue strength at $10^5$ cycles $\sigma_{1E5}$ that are used within the equation of the S–N curve, Equation (35), can be found in Table A5, where the bias-corrected log-standard deviations $s_{\log,N}$ [64] for the test series in the direction of fatigue life are also provided.

**Table A1.** Test results for pure bending tests.

| Amplitude of | | No. of Cycles until Crack Initiation | Remark |
| Bending Moment $M_{b,a}$/Nmm | Torsional Moment $M_{t,a}$/Nmm | | |
|---|---|---|---|
| 528,686 | 0 | 14,440 | |
| 447,213 | 0 | 38,830 | |
| 400,000 | 0 | 35,000 | |
| 378,297 | 0 | 80,000 | |
| 357,771 | 0 | 100,000 | |
| 338,359 | 0 | 165,000 | |
| 320,000 | 0 | 187,000 | |
| 302,638 | 0 | 550,000 | |
| 286,216 | 0 | 2,000,000 | no failure |
| 240,000 | 0 | 2,000,000 | no failure |

**Table A2.** Test results for pure torsion tests.

| Amplitude of | | No. of Cycles until Crack Initiation | Remark |
| Bending Moment $M_{b,a}$/Nmm | Torsional Moment $M_{t,a}$/Nmm | | |
|---|---|---|---|
| 0 | 940,000 | 5000 | |
| 0 | 881,250 | 13,900 | |
| 0 | 822,500 | 18,800 | |
| 0 | 763,750 | 32,000 | |
| 0 | 705,000 | 60,000 | |
| 0 | 705,000 | 77,000 | |
| 0 | 646,250 | 142,000 | |
| 0 | 646,250 | 312,000 | |

**Table A3.** Test results for proportional combination of bending and torsion ($\tau_{xy,a}/\sigma_{x,a} = 0.75$).

| Amplitude of | | No. of Cycles Until Crack Initiation | Remark |
| Bending Moment $M_{b,a}$/Nmm | Torsional Moment $M_{t,a}$/Nmm | | |
|---|---|---|---|
| 480,000 | 856,595 | 1220 | |
| 440,000 | 785,212 | 2300 | |
| 400,000 | 713,829 | 5100 | |
| 320,000 | 571,063 | 31,000 | |
| 280,000 | 499,680 | 76,000 | |
| 240,000 | 428,297 | 428,000 | |

**Table A4.** Test results for nonproportional combination of bending and torsion ($\tau_{xy,a}/\sigma_{x,a} = 0.75$), phase shift between torsion and bending $\Delta\varphi = 90°$.

| Amplitude of | | No. of Cycles until Crack Initiation | Remark |
| Bending Moment $M_{b,a}$/Nmm | Torsional Moment $M_{t,a}$/Nmm | | |
|---|---|---|---|
| 480,000 | 856,575 | 910 | |
| 400,000 | 713,930 | 2700 | |
| 360,000 | 642,255 | 7300 | |
| 340,000 | 606,770 | 12,800 | |
| 320,000 | 571,050 | 11,000 | |
| 300,000 | 535,330 | 65,200 | |
| 280,000 | 499,610 | 53,000 | |
| 260,000 | 463,890 | 157,000 | |

**Table A5.** Parameters describing the S–N curve.

| Test Series | Fatigue Strength at $10^5$ Cycles $\sigma_{1E5}$/MPa | Inclination k | Standard Deviation $S_{log,N}$ |
|---|---|---|---|
| pure bending | 405 | 5.9 | 0.184 |
| pure torsion | 310 ($\tau_{1E5}$) | 9.3 | 0.142 |
| proportional | 307 | 8.3 | 0.095 |
| nonproportional | 293 | 8.4 | 0.145 |

*Appendix A.3. Strain-Controlled Fatigue Test on Unnotched Specimens*

The pure axial tests were conducted on a servohydraulic testing machine with a load frame from MTS and an actuator from Schenck with a nominal force of 20 kN. The pure axial tests were conducted on a servohydraulic testing machine that can apply both tension and torsion to the specimens (Inova FUO 160-1260 M01). A multiaxial extensometer (MTS 632.80C-04) with a gage length of 25 mm was used to measure the axial and torsional strain (Figure A3).

Both the strain amplitudes that were imposed on the specimens during the individual tests and the resulting stabilized stress amplitudes are given in Tables A6–A9.

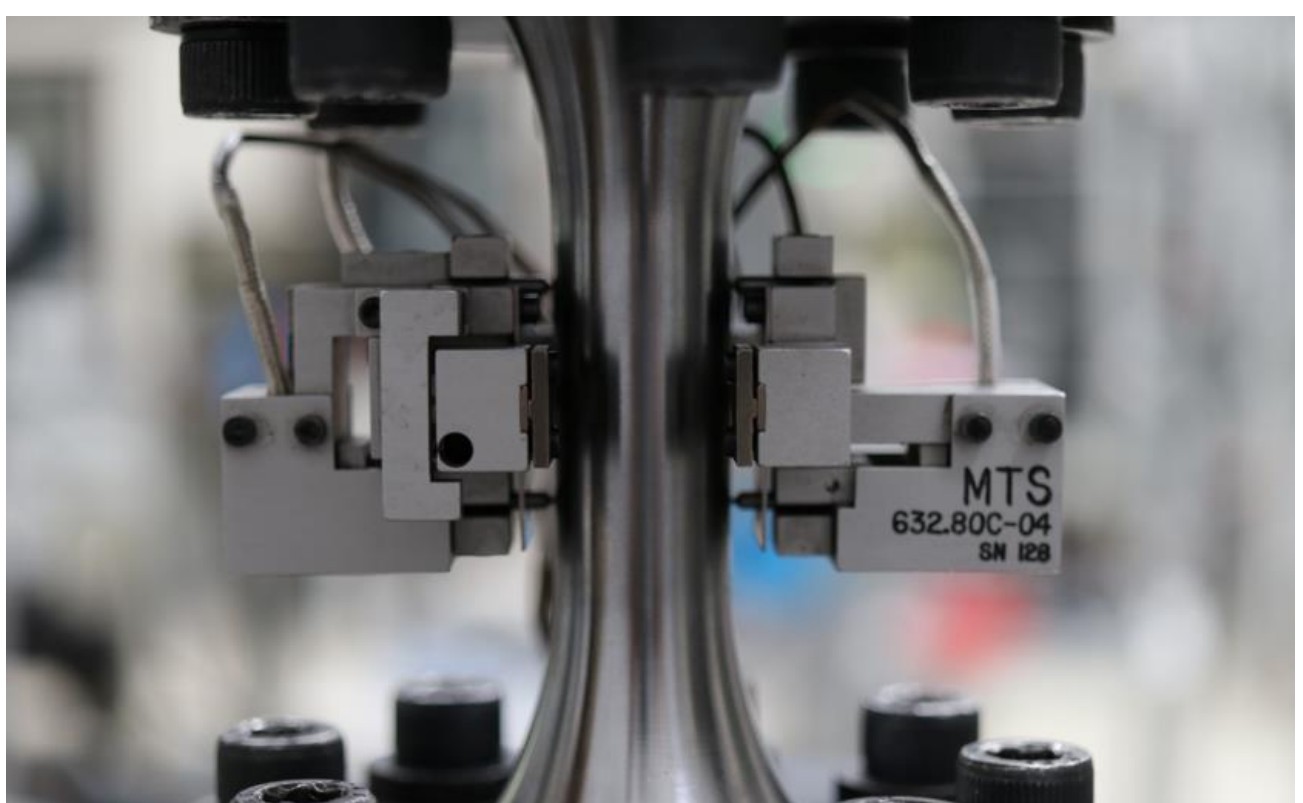

**Figure A3.** Multiaxial extensometer.

**Table A6.** Test results for pure axial tests.

| Axial Strain Amplitude $\varepsilon_a$/% | Torsional Strain Amplitude $\gamma_a$/% | Axial Stress Amplitude $\sigma_a$/MPa | Torsional Stress Amplitude $\tau_a$/MPa | Remark |
|---|---|---|---|---|
| 0.40 | 0 | 464.4 | 0 | |
| 0.40 | 0 | 486.6 | 0 | |
| 0.80 | 0 | 539.5 | 0 | |
| 0.20 | 0 | 303.7 | 0 | |
| 0.40 | 0 | 484.3 | 0 | |
| 0.80 | 0 | 539.3 | 0 | |
| 0.20 | 0 | 335.5 | 0 | |
| 0.40 | 0 | 486.4 | 0 | |
| 0.80 | 0 | 542.2 | 0 | |
| 0.20 | 0 | 332.9 | 0 | |
| 0.18 | 0 | 306.4 | 0 | |
| 0.16 | 0 | 277.1 | 0 | |
| 0.17 | 0 | 288.5 | 0 | |
| 1.20 | 0 | 566.5 | 0 | |
| 1.60 | 0 | 586.0 | 0 | |

**Table A7.** Test results for pure torsional tests.

| Axial Strain Amplitude $\varepsilon_a$/% | Torsional Strain Amplitude $\gamma_a$/% | Axial Stress Amplitude $\sigma_a$/MPa | Torsional Stress Amplitude $\tau_a$/MPa | Remark |
|---|---|---|---|---|
| 0 | 0.73 | 0 | 316.7 | |
| 0 | 0.88 | 0 | 297.6 | |
| 0 | 0.44 | 0 | 275.9 | |
| 0 | 1.03 | 0 | 320.0 | |
| 0 | 1.17 | 0 | 319.1 | |
| 0 | 0.40 | 0 | 266.6 | |
| 0 | 0.37 | 0 | 245.6 | |
| 0 | 1.32 | 0 | 324.6 | |
| 0 | 2.00 | 0 | 314.9 | |
| 0 | 1.90 | 0 | 319.6 | |
| 0 | 2.20 | 0 | 355.3 | |
| 0 | 0.60 | 0 | 315.6 | |

**Table A8.** Test results for proportional tests ($\gamma_a / \varepsilon_a = 1.9$).

| Axial Strain Amplitude $\varepsilon_a$/% | Torsional Strain Amplitude $\gamma_a$/% | Axial Stress Amplitude $\sigma_a$/MPa | Torsional Stress Amplitude $\tau_a$/MPa | Remark |
|---|---|---|---|---|
| 0.2 | 0.38 | 295.4 | 212.6 | |
| 0.4 | 0.88 | 350.1 | 229.5 | |
| 0.6 | 1.14 | 340.3 | 222.1 | |
| 0.2 | 0.38 | 301.3 | 217.9 | |
| 0.5 | 0.95 | 335.5 | 219.5 | |

**Table A9.** Test results for nonproportional combination of axial and torsional load ($\gamma_a / \varepsilon_a = 1.9$), phase shift between torsion and bending $\Delta\varphi = 90°$.

| Axial Strain Amplitude $\varepsilon_a$/% | Torsional Strain Amplitude $\gamma_a$/% | Axial Stress Amplitude $\sigma_a$/MPa | Torsional Stress Amplitude $\tau_a$/MPa | Remark |
|---|---|---|---|---|
| 0.2 | 0.38 | 297.7 | 209.4 | |
| 0.4 | 0.76 | 472.9 | 303.0 | |
| 0.3 | 0.57 | 435.8 | 285.8 | |
| 0.35 | 0.67 | 454.2 | 294.8 | |
| 0.5 | 0.95 | 494.6 | 315.3 | |

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
