# Peer review of "On Scaled Normal Stresses in Multiaxial Fatigue and Their Exemplary Application to Ductile Cast Iron"

_2673-3161, doi:10.3390/applmech3010018_

Round 1

Reviewer 1 Report

The manuscript deals with the critical plane approach of scaled normal stresses as proposed by Gaier and Dannbauer and modified by Riess et al. Authors created a deep overview in the field, which is beneficial to reader when aiming to apply it. As an example serves own experimental data of ductile cast iron material EN‐GJS‐500‐14. Approaches mentioned above give nonconservative predictions for the nonproportional loading cases. That is why there are proposed two modifications to improve the prediction under nonproportional loading. The first one is based on a notch root approximation according to Neuber with a different cyclic strain hardening exponents for proportional and nonproportional loadings. However, the prediction results are not improved significantly. The second correction is based on the Gaier’s shifting of the S–N curve for proportional loading. The reduction in fatigue strength of the S–N curve for proportional loading is set to 75% for the ductile cast iron material. The reduction factor is obtained as a best fit between nonproportional and proportional loading results. As discussed by authors too, for practical applications, a formulation of the nonproportionality measure would be necessary. The paper is well written and brings new findings on the basis of phenomenological approach. I recommend to publish it after major revisions.

My suggestions:

  • I recommend to extent the Introduction by approaches, which do not require rainflow methods application, for instance:

Jiang. Y. A fatigue criterion for general multiaxial loading. Fatigue Fract. Eng. Mater. Struct. 23, 19–32 (2000)

Volkov, I.A., Igumnov, L.A., Dell’isola, F., Litvinchuk, S.Y., Eremeyev, V.A.: A continual model of a damaged medium used for analyzing fatigue life of polycrystalline structural alloys under thermal-mechanical loading. Continuum Mech. Thermodyn. 32, 229–245 (2020)

and recent applications of the approaches.

  • In the section 5.3, the description of Neuber’s-like method is not sufficient. Authors should explain it in more details or redirect the reader to their published works, where the approach is described more deeply.
  • Please, be precise in citations of scientific works by numbered references (e.g. Gaier et al. –line 815, there are two papers, which one is considered?, line 512, etc)
  • After eq. (24), a description of the epsilon_ap (nonproportional) should follow. Is it a radius of the circumscribed circle?
  • It is not clear, how the correction by Gaier’s shifting of the S–N curve for proportional loading is done. Is there any connection to equation (35)? Usually, the critical plane criterion is used in the form: Fatigue Parameter = f (N). It can help to reader to see such a form of equation with implemented fnp.

Author Response

Reviewer 1: The manuscript deals with the critical plane approach of scaled normal stresses as proposed by Gaier and Dannbauer and modified by Riess et al. Authors created a deep overview in the field, which is beneficial to reader when aiming to apply it. As an example serves own experimental data of ductile cast iron material EN-GJS-500-14. Approaches mentioned above give nonconservative predictions for the nonproportional loading cases. That is why there are proposed two modifications to improve the prediction under nonproportional loading. The first one is based on a notch root approximation according to Neuber with a different cyclic strain hardening exponents for proportional and nonproportional loadings. However, the prediction results are not improved significantly. The second correction is based on the Gaier’s shifting of the S–N curve for proportional loading. The reduction in fatigue strength of the S–N curve for proportional loading is set to 75% for the ductile cast iron material. The reduction factor is obtained as a best fit between nonproportional and proportional loading results. As discussed by authors too, for practical applications, a formulation of the nonproportionality measure would be necessary.
The paper is well written and brings new findings on the basis of phenomenological approach. I recommend to publish it after major revisions.

Authors: Thank you for this positive evaluation of our paper and the effort involved. We have studied your comments with great interest and adjusted the paper in the according ways. This will surely help to improve the paper. Thank you very much for your review!

All references to equations and lines are referenfing to the updated version of the paper and deactivated

Reviewer 1: I recommend to extent the Introduction by approaches, which do not require rainflow methods application, for instance:

Jiang. Y. A fatigue criterion for general multiaxial loading. Fatigue Fract. Eng. Mater. Struct. 23, 19–32 (2000)

Volkov, I.A., Igumnov, L.A., Dell’isola, F., Litvinchuk, S.Y., Eremeyev, V.A.: A continual model of a damaged medium used for analyzing fatigue life of polycrystalline structural alloys under thermal-mechanical loading. Continuum Mech. Thermodyn. 32, 229–245 (2020)

and recent applications of the approaches.

Authors: We added these alternative approaches in section 2 (lines 127-141).

Reviewer 1: In the section 5.3, the description of Neuber’s-like method is not sufficient. Authors should explain it in more details or redirect the reader to their published works, where the approach is described more deeply.

Authors: We detailed the explanation of this correction approach (lines 801-836) and also added some more references to literature on the topic of Neuber’s method in fatigue calculation.

Reviewer 1: Please, be precise in citations of scientific works by numbered references (e.g. Gaier et al. –line 815, there are two papers, which one is considered?, line 512, etc)

Authors: The confusion about the different references of “Gaier et al.” is well understandable. We apologize. We now added the precise references throughout the text.

Reviewer 1: After eq. (24), a description of the epsilon_ap (nonproportional) should follow. Is it a radius of the circumscribed circle?

Authors: We are not absolutely sure what you mean by “circumscribed circle” but we added more explanation and references on epsilon_ap (nonproportional) and its determination, cf. lines 427-434.

Reviewer 1: It is not clear, how the correction by Gaier’s shifting of the S–N curve for proportional loading is done. Is there any connection to equation (35)? Usually, the critical plane criterion is used in the form: Fatigue Parameter = f (N). It can help to reader to see such a form of equation with implemented f.

Authors: We added the alternative formulation of the S-N curve for proportional loading (Eq. 35) as sig_a = f(N) and also integrated the reduction factor for nonproportional loading into Eq. 41 based on this new version of Eg. 35.

Reviewer 2 Report

Dear Authors,

Thank you for submitting your paper to Applied Mechanics. In the paper, the critical plane approach proposed by Gaier and Dannbauer and Riess et al. is presented in detail.  After opening the article I was surprised by its size (36 pages), but it turned out that it makes good reading. In fact, I have read it with great interest and instead of critical remarks, I marked in the text parts important for me. The paper is complete. The introduction initially considered too long is necessary to explain the fundamentals of the new approaches. I cannot really find a weak point in the article that requires any corrections.

Best regards,

Author Response

Reviewer 2: Thank you for submitting your paper to Applied Mechanics. In the paper, the critical plane approach proposed by Gaier and Dannbauer and Riess et al. is presented in detail. After opening the article I was surprised by its size (36 pages), but it turned out that it makes good reading. In fact, I have read it with great interest and instead of critical remarks, I marked in the text parts important for me. The paper is complete. The introduction initially considered too long is necessary to explain the fundamentals of the new approaches. I cannot really find a weak point in the article that requires any corrections.

Authors: Thank you very much for this excellent evaluation of our paper and the effort involved - especially since the paper has the page count mentioned above.

Round 2

Reviewer 1 Report

All my recommendations were taken into account. Good work.